# Unzipping of black phosphorus to form zigzag-phosphorene nanobelts

Zhifang Liu [1,6], Yilin Sun [2,6], Huaqiang Cao [1✉], Dan Xie [2✉], Wei Li[3], Jiaou Wang [4] & Anthony K. Cheetham [5✉]

Phosphorene, monolayer or few-layer black phosphorus, exhibits fascinating anisotropic properties and shows interesting semiconducting behavior. The synthesis of phosphorene nanosheets is still a hot topic, including the shaping of its two-dimensional structure into nanoribbons or nanobelts. Here we report electrochemical unzipping of single crystalline black phosphorus into zigzag-phosphorene nanobelts, as well as nanosheets and quantum dots, via an oxygen-driven mechanism. The experimental results agree well with our theoretical calculations. The calculation for the unzipping mechanism study suggests that interstitial oxygen-pairs are the critical intermediate species for generating zigzag-phosphorene nanobelts. Although phosphorene oxidation has been reported, lengthwise cutting is hitherto unreported. Our discovery of phosphorene cut upon oxidation represents a previously unknown mechanism for the formation of various dimensions of phosphorene nanostructures, especially zigzag-phosphorene nanobelts. It opens up a way for studying the quantum effects and electronic properties of zigzag-phosphorene nanobelts.

[1] Department of Chemistry, Tsinghua University, Beijing 100084, China. [2] Institute of Microelectronics, Beijing National Research Center for Information Science and Technology (BNRist), Tsinghua University, Beijing 100084, China. [3] Center of Rare Earth and Inorganic Functional Materials, National Institute for Advanced Materials, Nankai University, Tianjin 300350, China. [4] Beijing Synchrotron Radiation Facility, Institute of High Energy Physics, Chinese Academy of Sciences, Beijing 100049, China. [5] Department of Materials Science and Metallurgy, University of Cambridge, Cambridge CB3 0FS, UK. [6] These authors contributed equally: Zhifang Liu, Yilin Sun. ✉email: hqcao@mail.tsinghua.edu.cn; xiedan@tsinghua.edu.cn; akc30@cam.ac.uk

Phosphorene, which is a monolayer of black phosphorus (BP) in the strictest term, or few-layer (<10 layers) form of BP in a broad sense[1–3], has a natural bandgap, unlike graphene, and has aroused great interest[1,4–18]. BP can be synthesized by different methods[1,19,20], from original methods involving high pressure and high temperature from white and red phosphorus, followed by mercury as a catalyst and the bismuth-flux methods, to the current low-pressure transport reaction routes. However, the effective preparation method of phosphorene is a key factor limiting its application. Top-down exfoliation methods, including mechanical[4] and liquid[14–16] exfoliation techniques, have been applied to generate phosphorene, while bottom-up phosphorene synthesis techniques have not yet been realized. Only small-size phosphorene can be generated by the mechanical "scotch-tape" method, which cannot be scale up. Reported liquid exfoliation, consisting of ultrasonic exfoliation of bulk BP immersed into a solvent, is more suitable for the preparation of phosphorene in scale up production[15]. Other techniques, such as pulsed laser deposition to prepare amorphous BP ultrathin films[17], and plasma thinning technique after mechanical cleavage to prepare monolayer phosphorene[18], have also been developed to fabricate phosphorene from bulk BP as the precursor. Also electrochemical molecular intercalation approach has been applied to produce monolayer phosphorene molecular superlattices consisting of alternating layers of monolayer two-dimensional (2D) phosphorene atomic crystals and molecular layers[13]. Obtaining 2D nanostructured phosphorene is still a great challenge, let alone shaping its 2D structure as nanobelts or nanoribbons[21]. The production of nanobelts or nanoribbons of any materials is a hot topic, because it would constitute a basis for developing one-dimensional nanoelectronics[22]. Confirmed by theory studies, the bandgap size and the effective mass of charged carriers in phosphorene nanoribbons (PNRs) present very sensitive to ribbon width and crystallographic orientation, due to their strong anisotropic in-plane and interplanar properties[6,23–25]. Watts et al. recently reported a method for creating PNRs by ionic scissoring macroscopic BP crystals[5]. This method follows a two-step process, i.e., bulk BP intercalated with lithium ions, followed by being immersed in an aprotic solvent and mechanically agitated. The synthesis method can yield predominately single-layer thickness PNRs with widths of 4–50 nm and aspect ratios of up to 1000, but it is time-consuming, low-temperature treatment and the program is complex.

Here, we propose a facile method of electrochemical exfoliation to synthesize not only 2D phosphorene nanosheets (Supplementary Fig. 1a, b), but also zigzag-phosphorene nanobelts (z-PNBs) (Fig. 1 and Supplementary Fig. 1c) and phosphorene quantum dots[26] (Supplementary Fig. 1d, e), only by changing the current densities. Our method for z-PNBs production follows a two-step process, i.e., bulk BP crystals are intercalated with $BF_4^-$ ions, followed by an oxygen-driven unzipping mechanism. The process is illustrated in Fig. 2 and supported by calculation (Fig. 3). This method for generating z-PNBs paves a route to constructing electronic devices and studying the electronic properties of phosphorene (Fig. 4).

## Results and discussion

**Structural characterization**. It is well known that the structure of orthorhombic BP with lattice constants of $a = 3.314$ Å, $b = 10.473$ Å, $c = 4.374$ Å[27–29], consists of puckered layers parallel with the (010) plane stacked along the [010] direction with the layer-to-layer spacing of 0.53 nm[10,24,30], as shown in Fig. 2a. It is worth pointing out that the Raman mode attribution directly depends on the selection of the crystal axis. Each phosphorene layer is composed of six-membered P rings in chair configuration arranged in

a puckered honeycomb structure, and pairs of the six-membered P rings are structured in a cis-decalin manner. Therefore, the zigzag (ZZ) direction and armchair (AC) direction are orthogonal. Due to the $sp^3$ hybridization of phosphorus atoms[27,31], which is different from other 2D layered materials, the primitive cell contains four atoms, whose size is a half of the unit cell of BP (i.e., $b/2 = 5.237$ Å)[28,32]. As a result of the puckered structure, each single layer contains two atomic layers and two kinds of P–P bonds, i.e., shorter P–P bond length of $d_1 = 0.2224$ nm connecting the nearest P atoms in the same plane, and longer P–P bond length of $d_2 = 0.2244$ nm connecting P atoms between the top and bottom of a single layer; and two kinds of P–P–P bond angles, i.e., $\theta_1 = 96.34°$ connecting three P atoms in the same plane, and $\theta_2 = 102.09°$ connecting three P atoms between the top and bottom of a single layer (Fig. 2a)[28,31,32]. Obviously, longer P–P bonds with $d_2$ bond length are more easily broken than shorter P–P bonds with $d_1$ bond length, which means that longer P–P bonds are highly active compared with shorter P–P bonds[2,28].

Different P–P bond lengths in two orthogonal directions, known as ZZ and AC, result in the anisotropic properties of phosphorene[24]. It is also demonstrated that the bandgap and Young's modulus of zigzag PNRs (z-PNRs) are larger than those of armchair PNRs (a-PNRs)[11,23]. These exotic properties along the ZZ lattice direction are predicted to overstep that of a-PNRs in thermal conductivity[1], semiconductor behavior[23], and mechanical strength[11], leading to the great potential of z-PNRs for a broad range of applications such as thermoelectric devices, flexible electronics, and quantum information technologies[5], etc.

Based on the theory that BP oxidation can lead to BP degradation, we designed an electrochemical method to controllably prepare phosphorene by controlling the oxygen concentration, i.e., unzipping bulk BP to nanosheets, nanobelts, and quantum dots through controllable oxidation at room temperature (Fig. 1 and Supplementary Figs 1, 2). During this process, bulk BP (anode) and Pt flake (cathode) were immersed into 1-methyl-3-butylimidazolium tetrafluoroborate ([BMIM]$BF_4$)/water solution with weight ratio of 1:2. When the power is turned on, the $BF_4^-$ anions move toward the anode (BP) and are inserted between BP layers, which results in the expansion of the layer spacing of the BP crystals; and the more oxygen formed at anode due to electrolysis of water will accelerate the unzipping P–P bonds of BP.

Figure 1a shows a transmission electron microscopy (TEM) image of a nanobelt with its edges along ZZ direction (i.e., [100] direction) of phosphorene. The observed wrinkle plate indicates that the nanobelt is very thin. High-resolution TEM (HRTEM) image of nanobelt (Fig. 1b), presenting the (200) crystal plane spacing of 0.17 nm[27–29], indicates that the zone axis is [010]. Based on the interplanar spacing $d_{hkl}$ of orthorhombic system[33] and the reported unit-cell dimensions of BP[27–29], we can obtain the calculated $d_{200} = a/2 \approx 0.166$ nm. And the measured lattice fringes with a spacing of $d_{200} = 0.167$ nm, with error of 0.60% or 0.79%, compared with calculated $d_{200} = 1.66$ Å and the X-ray diffraction (XRD) card value (JCPDS No. 73-1358) of $d_{200} = 1.6568$ Å, respectively. The crystal planes (200) are perpendicular to the zone axis [010] and parallel to the a, c-plane.

The selected-area electron diffraction (SAED) pattern also demonstrates the crystal zone axis is along [010] direction (i.e., out-of-plane direction) (Fig. 1c), which is perpendicular to the plane (200) in orthorhombic systems. Theoretically indicated that the intensity ratio of the (101) and (200) diffraction spots [$I_{(101)}$/$I_{(200)}$] is thickness dependent[34]. Therefore, according to the $I_{(101)}$/$I_{(200)}$ value of 0.292, we attribute the nanobelt to trilayer phosphorene (inset of Fig. 1c)[34]. Figure 1d shows a twisted morphology nanobelt, indicating that the z-PNBs own excellent flexibility.

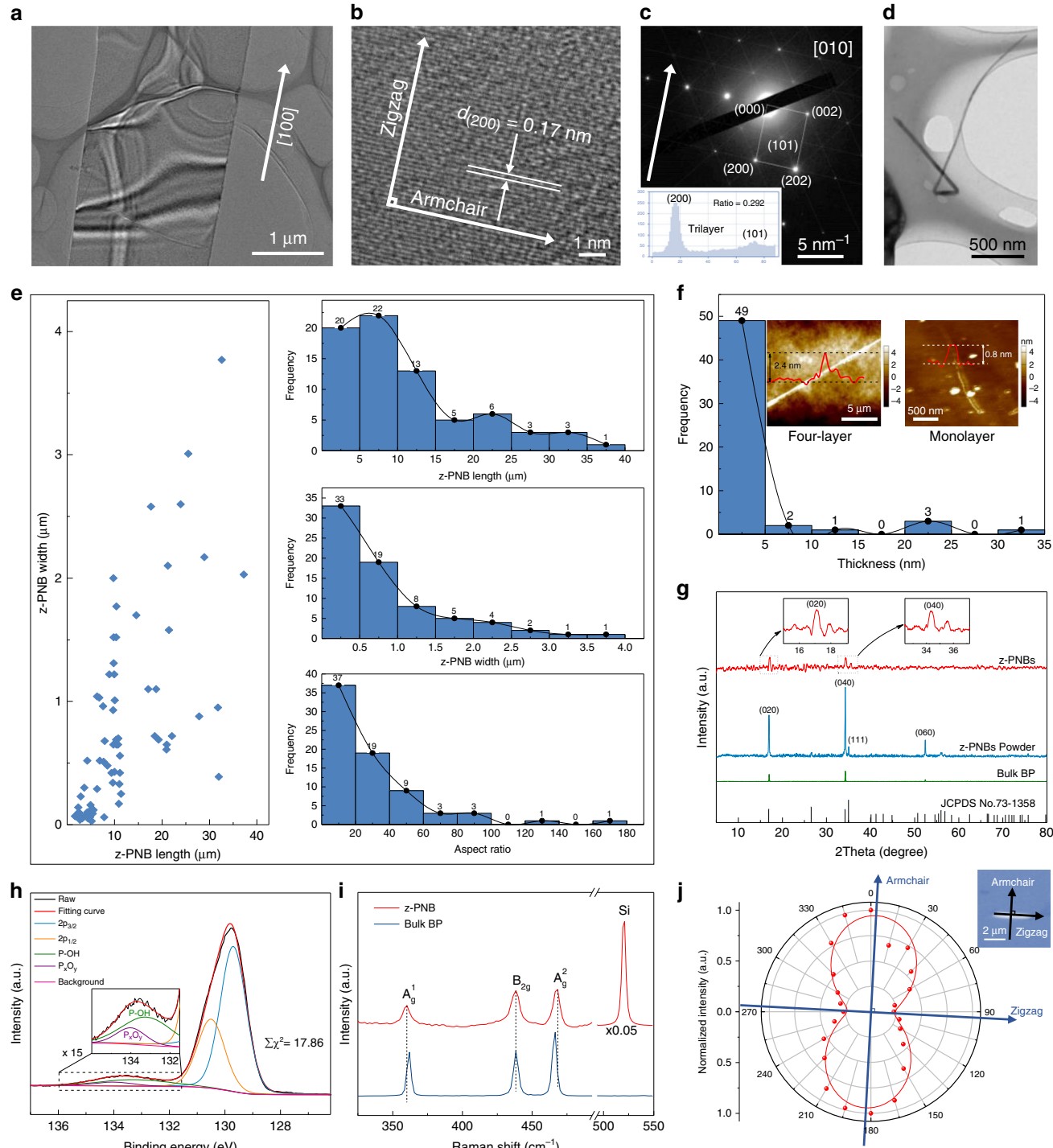

**Fig. 1 Characterizations of z-PNBs. a** TEM image of a single z-PNB with the zigzag directions, corresponding to the [100] direction. **b** Corresponding HRTEM image of **a**. **c** Corresponding SAED pattern of **b**, the zigzag and armchair directions are shown by white arrows. Inset: intensity of (200) and (101) fast Fourier transform (FFT) patterns. **d** TEM image of an individual twisted z-PNBs. **e** Size distribution diagram of the TEM images statistics on 73 z-PNBs. Left, diagrams of length as a function of width for 73 z-PNBs; right, frequency distribution of z-PNBs. length (top), width (middle), and aspect ratio (bottom). **f** Thickness distribution diagram from the AFM images of 56 z-PNBs. Inset: AFM image of two typical belts onto a 300 nm $SiO_2$/Si substrate with the thickness of ~2.4 nm (left) and ~0.8 nm (right), corresponding to four-layer and monolayer phosphorene, respectively. **g** XRD patterns of as-prepared z-PNBs. Inset: ten times amplification of XRD patterns in dashed rectangle areas of z-PNBs. **h** High-resolution XPS spectra of the P $2p$ signal for electrochemically exfoliated z-PNBs with $\sum X^2$ of 17.86. Inset: 15 times amplification of XPS signals in dashed rectangles. **i** Raman spectra ($\lambda = 532$ nm) of z-PNBs and bulk black phosphorus on $SiO_2$/Si substrates with 300 nm in thickness, respectively. **j** Normalized $A_g^2$ Raman band of z-PNBs from polarization-resolved Raman spectra of a z-PNB (Supplementary Fig. 5c) as a function of rotation angle. Red dots and lines are the experimental data of z-PNBs and the fitting curve by a $\sin^2\theta$ function. Top right: optical image of a z-PNB on $SiO_2$/Si substrates with 300 nm in thickness. The arrows indicate the identified armchair and zigzag directions, respectively.

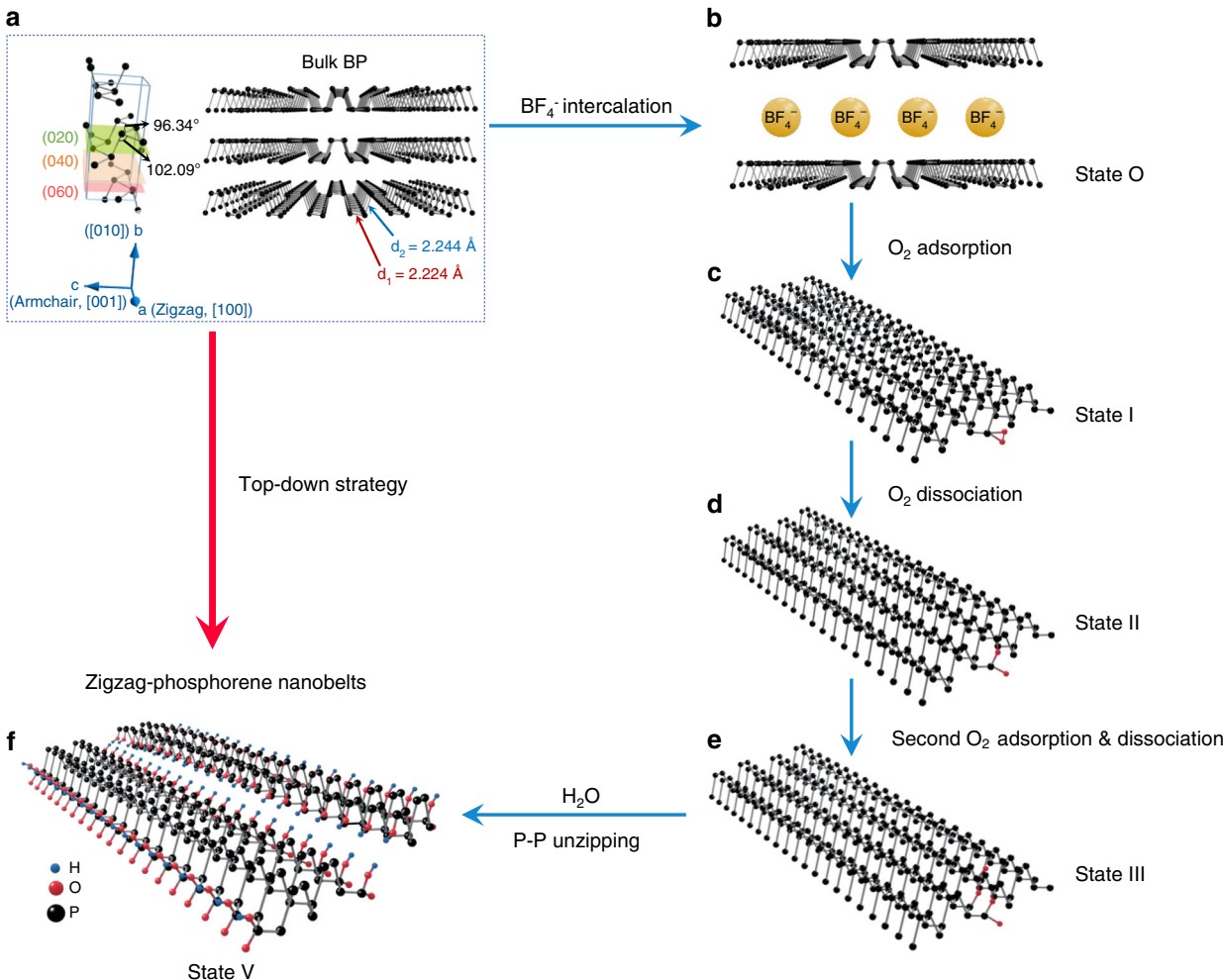

**Fig. 2 Mechanism of preparation of z-PNBs from bulk BP.** Step 1: interaction; **a** bulk BP was used as the raw material. **b** Bulk BP is intercalated with $BF_4^-$ into few-layer phosphene. Step 2: **c-f** oxygen-driven unzipping process of phosphene into z-PNBs at the presence of $H_2O$ molecules.

The size distribution of z-PNBs with different aspect ratios is shown in Fig. 1e and Supplementary Fig. 2f, indicating that the aspect ratio is generally larger than 10. The morphology and height of z-PNBs were further characterized by atomic force microscope (AFM). The statistical results indicate that the thicknesses of most of (close to ~90%) nanobelts is ~2.7 ± 1.7 nm, corresponding to ~1–8 layers (Fig. 1f and Supplementary Fig. 3), since the monolayer phosphene defined as synthesized in the liquid phase is $0.84\,nm^2$, thus the extra height could be calculated to be 0.31 nm compared with the 0.53 nm of theoretical thickness of monolayer phosphene. Two typical individual nanobelts with the thickness of ~2.4 and ~0.8 nm are shown as inset of Fig. 1f, corresponding to four-layer and monolayer phosphene, respectively. The observed tip effect observed in the AFM images (Supplementary Fig. 3) might be caused by tip contaminations, not by broken tip, due to new AFM tip was used in the measurements.

XRD pattern (Fig. 1g) indicates the as-prepared nanobelts belong to orthorhombic phase of BP, with preferred peaks corresponding to (0k0) (k = 2, 4, 6) planes, revealing a highly oriented to the [010] direction of phosphene nanobelt, proving the layered nature[35].

X-ray photoelectron spectroscopy (XPS) survey spectrum presented P but no other purities except for C and O adsorbed onto the sample during unavoidable sample tests processes (Supplementary Fig. 4a). The high-resolution XPS spectrum of P

2p presents doublet characteristic peaks of crystalline BP at 129.7 eV of P $2p_{3/2}$ and 130.5 eV of P $2p_{1/2}$, respectively[36], as well as two small peaks at 133.2 eV of P–OH, 134.0 eV of oxidized phosphorus ($P_xO_y$) sub-band[37]. The percentage of P–OH among all P–O species was 74.26 at.%. The O content was then calculated to be 9.90 at.% from the high-resolution P 2p spectra of the z-PNBs (Fig. 1h and Supplementary Table 1), in agreement with that of energy dispersive spectrum (EDS) mapping (Supplementary Fig. 4c–e). The O/P ratio of 0.11 (Fig. 1h) was much lower than the limit of 0.89 for preventing degradation[38].

The structure information of the phosphene was further characterized by the Raman technique (Fig. 1i, j and Supplementary Fig. 5). Only three of six Raman active phonon modes can be detected when the incident laser is perpendicular to the phosphene plane[9]. In the spectra, four Raman peaks at 360.2, 438.3, 468.3, and 521.0 $cm^{-1}$ can be observed. The major scattering peak at 521.0 $cm^{-1}$ is from the Raman peak of the silicon substrate. The other three peaks are attributed to the atomic vibration along [010] crystallographic direction (i.e., b-axis direction, through-plane, TP direction) $A_g^1$ (out-of-plane mode), along [100] crystallographic direction (i.e., a-axis direction, ZZ direction) $B_{2g}$ (in-plane mode), and along [001] crystallographic direction (i.e., c-axis direction, AC direction) $A_g^2$ (in-plane mode) phonon modes of few-layer phosphene[27–29]. These observed Raman peaks of $A_g^1$, $B_{2g}$, $A_g^2$ phonon modes match

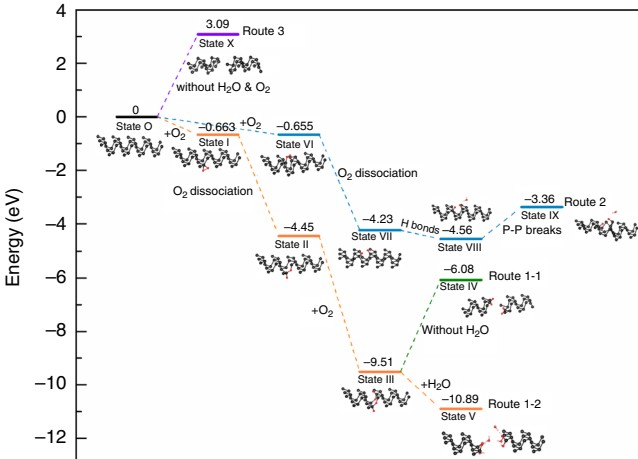

**Fig. 3 Reaction mechanism of the oxygen-driven unzipping black phosphorus processes.** Energy profile of the unzipping process through four possible routes: (i) Route 1-1, the formation of interstitial oxygen route under the conditions with $O_2$, but without $H_2O$ (green); (ii) Route 1-2, the formation of interstitial oxygen route under the conditions with $O_2$ and $H_2O$ (orange); (iii) Route 2, the formation of dangling oxygen route under the conditions with $O_2$ and $H_2O$ (blue); (iv) Route 3, through breaking P–P under the conditions without $H_2O$ and $O_2$ (purple), respectively. Corresponding side structures of each configuration for unzipping process are also shown inside.

well observed in bulk BP[27–29], suggesting that the electrochemical exfoliated phosphorene remained crystalline after the exfoliation. The value of $A_g^1/A_g^2$ is ~0.52 (Fig. 1i), indicating it is within the range of 0.4–0.6 for pristine phosphorene prepared in a glove box[30,39]. The intensity ratio of the $A_g^2$ and Si peaks (~0.01) was used to determine the thickness, corresponding to about bilayer phosphorene[34]. The $A_g^2$ mode shifts to higher frequency 468.3 cm$^{-1}$ of phosphorene nanobelts from 466.5 cm$^{-1}$ of bulk BP (Fig. 1i), due to the decreasing thickness, which demonstrates successful exfoliation of bulk BP[30,40]. This blueshift phenomenon of $A_g^2$ mode of phosphorene can be attributed to the long-range Coulombic interlayer interactions[40]. Also, the full-width at half-maximum of $A_g^2$ mode broadens from 3.52 cm$^{-1}$ for bulk phosphorus to 7.04 cm$^{-1}$ for phosphorene, close to that of bilayer phosphorene[30], due to the enhanced interaction with external environment compared with bulk phosphorus[40]. However, angle-resolved polarized Raman spectroscopy has been demonstrated to be used to nondestructively identify the special ZZ and AC crystalline direction of phosphorene, i.e, the relative largest or smallest $A_g^2$ mode intensity corresponds to the AC or ZZ direction of BP for an arbitrarily by rotating the arbitrarily located samples (Fig. 1j), respectively, due to the strong anisotropic nature of phosphorene itself[40]. When the Raman spectrum was measured following excitation laser polarization aligned close perpendicular to (here, a deviation of about 3°) the long edge direction of the phosphorene nanobelt (shown in the upper right inset of Fig. 1j), the smallest intensity of $A_g^2$ mode is observed (Fig. 1j). Thus, this phenomenon further demonstrates that the phosphorene nano-belts are along ZZ direction[40]. The reported ionic scissoring method[5] can generate high-quality, individual PNR; most (around 65%) are monolayer, but the synthesized steps are time-consuming and rigorous. By contrast, our electrochemical method has simple steps; and most (around 63%) of belts are <5 layers with a $PO_x$ content of ~9.9 at.% (measured after exposed to air for ~5 min during transfer to the XPS spectrometer). Sun

et al.[2] reported a liquid-phase exfoliation method to form phosphorene by immersing the BP into a sonic bath, followed by ultrasonically exfoliated. This method can form irregular-shaped few-layers phosphorene, because breaking P–P is random by a sonication method. However, as long as the current density is changed, our electrochemical method can generate not only z-PNBs, but also quantum dots and nanosheets.

**The z-PNB formation mechanism.** The formation process of z-PNBs can be divided by two steps, i.e., ionic intercalation and oxygen degradation (Fig. 2). The $BF_4^-$ ionic diameter is 0.436 nm (ionic radius, $r = 0.218$ nm)[41], which is smaller than the adjacent layer spacing of 0.53 nm for BP[24,30]. Thus $BF_4^-$ ions will be easily inserted between the phosphorene layers, along the $a$-axis oriented channels, i.e., [100] direction of BP[2]. The $BF_4^-$ inter-calation process can be manifested by the chronopotentiometry curves during the exfoliation process at different current densities ranging from ~0.1 to 0.5 A cm$^{-2}$ (Supplementary Fig. 1h). It was found that the voltage first drops rapidly, and then rises slowly. The decrease in voltage is due to the increase in surface area of BP, caused by $BF_4^-$ intercalation into the BP crystal, while the increase in voltage is due to the increase in the bandgap of phosphorene compared with that of BP crystal[4], caused by the intercalation of ions[2,42]. Furthermore, the intercalation inter-mediate step of the electrochemical synthesis method without ultrasonic treatment can be directly observed by the TEM observation (Supplementary Fig. 2d). At the same time of $BF_4^-$ intercalation, $O_2$ will be chemisorbed dissociatively on the surface of pristine BP, followed by formation of hydrogen bonds between dangling oxygen and water molecules; as well as the electronic hydrolysis of P–O–P. These processes lead to the P–P unzipped, which is supported by the theoretical calculation (Fig. 3 and Supplementary Figs. 6–8). It is generally believed that liquid exfoliation may introduce some vacancies on the surface of BP sheet[43]. Both the edges and vacancies are less stable, which are easily bonded with oxygen adatoms than the pristine surface of BP sheets[43]. The calculations and experiments show that $O_2$ is chemisorbed dissociatively on the surface of pristine BP, while $H_2O$ is physical adsorption[44]. At the same time with the inter-calation of $BF_4^-$ into BP, the oxygen molecules also chemisorb randomly onto the phosphorene layer surface or edges of the BP, which is an exoenergetic process, leading to the formation of neutral defects, i.e., dangling oxygen which can increase the hydrophilicity of phosphorene layers, supported by the calcula-tion (Fig. 3). This is driven by the existence of lone-pair electrons in the P atom on the surface of BP, which makes phosphorus very reactive to air[7]. Besides the oxygen chemisorption onto phos-phorene layer surface or edge, hydrogen bond will be formed between dangling oxygen and water molecules, which is an energy favorable process. Besides the most stable dangling oxy-gen, the metastable interstitial oxygen atoms, in which oxygen penetrates into the lattice and form a P–O–P bridge structure, and other metastable bridge-type surface oxygen defects including diagonal oxygen bridge (i.e., connecting P atoms on different edges of the ZZ ridges), and horizontal oxygen bridge (i.e., con-necting P atoms on the same edges of the ZZ ridges) can be formed (Supplementary Fig. 7)[7]. The metastable interstitial oxy-gen bridge and diagonal bridge oxygen configurations can be changed into to dangling oxygen via the spin-allowed phonon-mediated dissociation with an energy gain of 0.44 or 0.92 eV, respectively (Supplementary Fig. 7)[7]. The dangling oxygen atoms function as anchors for $H_2O$ by forming hydrogen bonds with $H_2O$ molecules. Although oxygen bridge-type defects are meta-stable, they can still be generated under nonequilibrium condi-tions[7]. Because some vacancies on the surface of BP can be

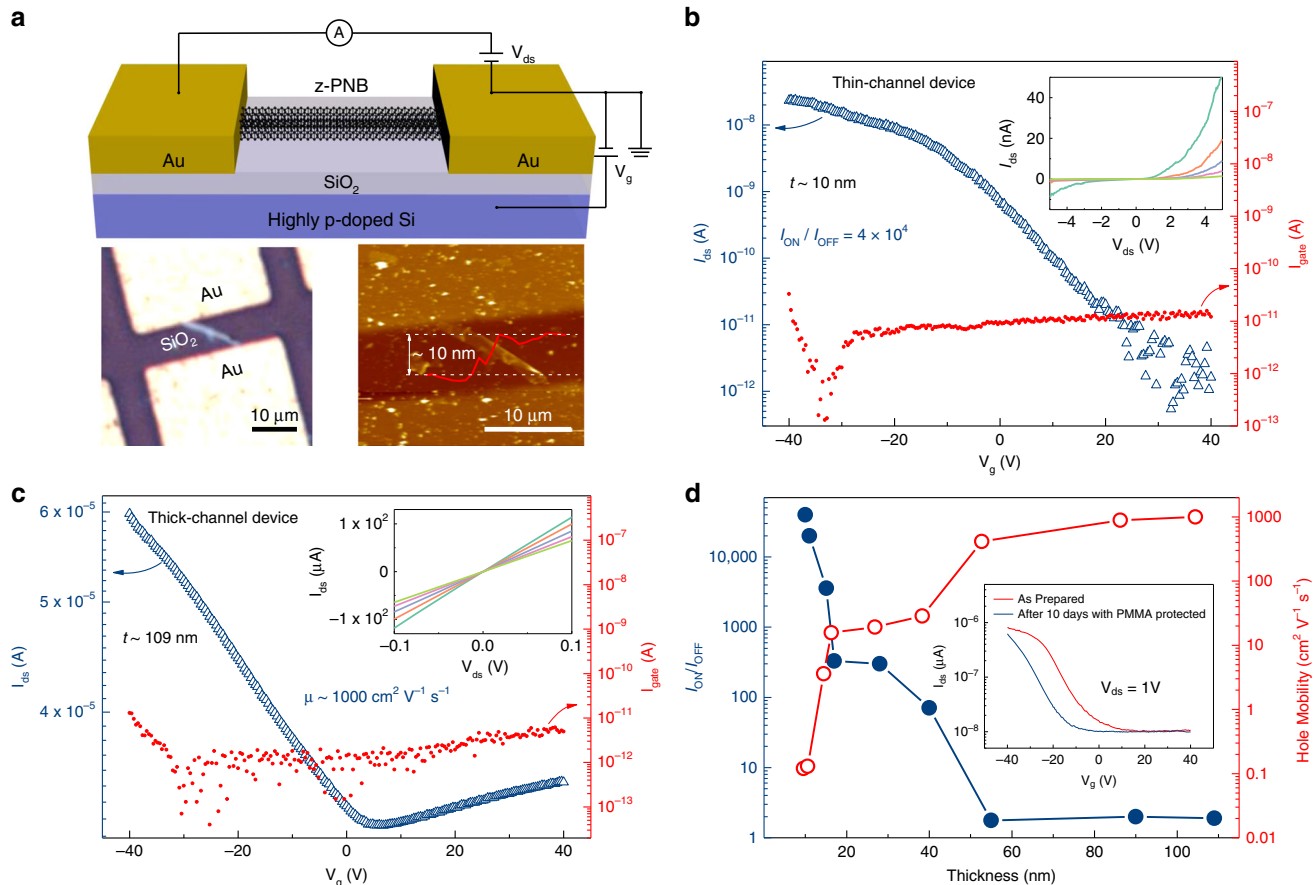

**Fig. 4 Characterization of individual z-PNB devices. a** Top: schematic view of z-PNB devices. Bottom: optical image (left side) and AFM image (right side) of a 10-nm z-PNB device. **b** Transfer characteristics of z-PNB device (on a logarithmic scale, left axis) with drain–source voltages of 3 V, $I_{ds}$, $I_{gate}$, $V_g$, $V_{ds}$ are the source–drain current, gate leakage current, back gate bias, and drain to source bias, respectively. The leakage currents of both devices were plotted as a function of $V_g$ (right axis) and found to be ~$10^{-11}$ A. Inset: output curves ($I_{ds} - V_{ds}$) with $V_g$ decreasing from −40 to 0 V at a step of 10 V. **c** Transfer characteristics of 109-nm z-PNB device with drain–source voltages of 0.05 V. The leakage current was found to be ~$10^{-11}$ A. Inset: output curves ($I_{ds} - V_{ds}$) with $V_g$ decreasing from −40 to 0 V at a step of 10 V. **d** $I_{ON}/I_{OFF}$ and hole mobility were extracted from different z-PNB devices as a function of the thickness of z-PNBs. Blue solid circle and red hollow circle represent the $I_{ON}/I_{OFF}$ and hole mobility, respectively. Note that all the electronic measurements were conducted at 300 K under $N_2$ atmosphere. Inset: stability of z-PNB-based device before and after 10 days with PMMA protected. The ON/OFF ratio and hole mobility remained stable with the protection of PMMA after 10 days.

introduced in liquid exfoliation process, the chemisorption of $O_2$ in adjacent vacancy is easier to adsorb than at the pristine sites of BP[43]. Therefore, the longer P–P bond ($d_2 = 2.244$ Å) in BP is more easily broken than the shorter P–P bond ($d_1 = 2.224$ Å) after oxidation by $O_2$ and subsequent interaction between oxygen defects and water molecules, supported by the calculation results (Fig. 3). Thus, it leads to a clean-cut edge along the [100] direction (i.e., ZZ direction), while it is impossible to obtain clean-cut edge along the [001] direction (i.e., AC direction), because the shorter P–P bonds will hold the phosphorene layer together[2]. Thus, we can obtain z-PNBs.

In addition, the morphology of phosphorene was found to be directly affected by adjusting the applied current densities. If we increase or decrease the current density of the electrochemical exfoliation of BP, which accelerates or decreases the oxidation rate accordingly, we can also obtain phosphorene quantum dots[26] or phosphorene nanosheets. This is due to the $O_2$ concentrations that are directly proportional to the current density (Supplementary Fig. 1g). It was worth nothing that low current density referred to a slow migration rate of $BF_4^-$ into the interlayer of BP and suppressed the oxygen concentration (Supplementary Fig. 1g). This led to thick phosphorene nanosheets. Oxygen-driven process can also enlarge the bandgap of z-PNBs, which

was further confirmed by ultraviolet–visible (UV–vis) absorption spectra and chronopotentiometry tests (Supplementary Fig. 1h, i). Also the BP precursor crystal quality will affect the electrochemical exfoliation processes, because defects are favorable sites for the oxygen adsorption.

**Theoretical calculations.** For further understanding the unzipping process of single crystalline BP to form z-PNBs, we carried out density functional theory (DFT) calculations with the Vienna Ab initio Simulation Package (VASP) using the Perdew–Burke–Ernzerhof (PBE) exchange-correlation functional (Fig. 3 and Supplementary Figs. 6–8). In order to explain the mechanism by calculation, we first define the interaction energy ($\Delta E$) for each configuration as follow: $\Delta E = E(\text{complex}) - [E(\text{surfactant}) + E(\text{surface})]$, where $E(\text{complex})$, $E(\text{surfactant})$ and $E(\text{surface})$ are the total energies of optimized surface–surfactant complex, surfactant molecule, and surface cluster, respectively[45]. It has been demonstrated that O chemisorption is energetically favored, which can generate dangling oxygen, interstitial oxygen bridge configurations (Supplementary Fig. 7)[7]. In fact, all $P_xO_y$ structures can be identified to the two above-mentioned P–O motifs: the dangling and the interstitial oxygen motifs[46].

After oxygen chemisorption on BP (State Original, State O) to form epoxy three-membered ring structure ($-0.663$ eV, state I), followed by subsequent $O_2$ dissociation to form a dangling oxygen and an interstitial oxygen bridge configurations in which the oxygen occupying a position close to a P–P bond center, including the longer P–P bond ($d_2 = 2.244$ Å) ($-4.45$ eV, State II) or the shorter P–P bond ($d_1 = 2.224$ Å) ($-3.90$ eV). Obviously, the $O_2$ dissociate on phosphorene sheet is a spontaneously process at room temperature. The chemisorption of the second $O_2$ molecule is determined by the presence of local impurities or defects[7]. That means the corresponding oxidized P atoms could be easily attacked during further oxidation processes. The possible seven configurations for the second $O_2$ chemisorption and dissociation on the oxidized P atom are shown in Supplementary Fig. 8—State III-a–g, in which two configurations have relaxation phenomenon after optimization (State III-f and -g), due to pristine structure is not the minimum point of potential energy surfaces. It shows that the lowest energy configuration is to form interstitial oxygen-pair structure ($-9.51$ eV, State III-a) from State II accompanied with exothermic energy of 5.06 eV among the seven possible configurations (Supplementary Fig. 8—State III). The very large released energy of 5.06 eV suggests a substantial large reaction rate for the oxygen chemisorption to interstitial oxygen-pair reaction at room temperature. The further breaking P–P bonds will lead to a high energy state (State IV, $-6.08$ eV) accompanied with an endothermic energy of 3.43 eV, without consideration of the formation of hydrogen bonds between water and oxygen defects (Route 1-1). This calculation indicates that the oxidation alone does not break down the P–P bonds of the phosphorene sheets, which was also demonstrated by reported literature (ref. [47]). However, followed by the interaction between defect oxygen and water molecules, i.e., the formation of hydrogen bonds between water molecules and dangling oxygen (P=O), and electrophilic hydrolysis of interstitial oxygen[48], it can lead to P–P breaking (a vacuum of ~17.8 Å was added in the direction normal to the monolayer, and another at least 12 Å vacuum separating the nanobelts after optimization, to avoid interactions, Supplementary Fig. 6) along the ZZ direction (i.e., along the direction for cutting interstitial oxygen-pair of P–O–P, to enter State V ($-10.89$ eV) accompanied with a large exothermic energy of 1.38 eV (Route 1-2), which is an energetically favored route. This calculation indicates that the P–P bonds can be break down easily along the ZZ direction via the interstitial oxygen-pair intermediate species when both $O_2$ and water are involved in the degradation process. After the breaking of P–P bonds at the position of the interstitial oxygen, the P–O–P will be changed into P–O–H. This calculation result is also supported by our experimental XPS results, that is the unzipped z-PNB products are terminated mostly with P–O–H located at 533.0 eV (74.26 at.%), dangling oxygen (P=O oxygen) located at 530.7 eV (14.04 at.%) and interstitial oxygen (P–O–P) located at 531.7 eV (11.70 at.%)[37], where the percentage of P–OH among all P–O species was 74.26 at.%. (Supplementary Fig. 4b and Supplementary Table 2), which is self-consistent with the P–O functional groups based on the peak of P $2p$ (Fig. 1h and Supplementary Table 1). And it is larger than that of the above-mentioned two intermediates, P–O–P and P=O, after $O_2$ dissociation (Fig. 2, State II), in which each should account for 50 at.%. That means most P–O–P configuration will be destroyed after the BP degradation, and changed into P–OH. If $O_2$ chemisorption ($-0.663$ eV, State I) is followed by dissociation, it will generate two interstitial oxygen P-O-P configurations (Supplementary Fig. 8-State II a and b), one P-O-P bond plus one short P-O-P configuration (Supplementary Fig. 8-State II c), and two short P-O-P bond configuration (Supplementary Fig. 8-State II d), respectively. It should be pointed out that the above two interstitial oxygen P–O–P configurations are not at lowest energy

points, which will lead to the atomic relaxation when the geometry is optimized. This indicates that the P–P breaking route via two interstitial oxygen P–O–P intermediate configurations is impossible. If $O_2$ chemisorption forms epoxy four-membered ring structure ($-0.655$ eV, State VI), it will subsequently dissociate to form two P=O dangling bond oxygens ($-4.23$ eV, Supplementary Fig. 8-State VII), due to the instability of epoxy four-membered ring structure. This kind dangling oxygen forms hydrogen bond with water molecules, which leads to State VIII ($-4.56$ eV). However, the energy barrier of 1.2 eV remains to be surmounted to break down the P–P bonds to enter State IX ($-3.36$ eV), after the formation of hydrogen bonds, suggesting that it is not an energy favorably process (Route 2) compared with Route 1-2. Under the conditions of without $O_2$ and water, the energy barrier for breaking P–P bond is 3.09 eV (State X), which is too high. This suggests that it is a strong endothermic process (Route 3). Our calculation for the unzipping mechanism study suggests that the interstitial oxygen-pair is a critical intermediate species. A previous DFT study indicates that H, OH, F, and Cl can act as scissors to cut phosphorene into nanoribbons or nanochains, while O, S, and Se atoms cannot[49]. In contrast, our unzipping mechanism shows that oxygen unzipping phosphorene is only available under the condition of the formation of hydrogen bonds with $H_2O$. This can explain the electrochemical exfoliation process in air with $H_2O$.

**Electronic performance.** Oxygen was unavoidably and irreversibly introduced during the exfoliation process, leading to some edge defects, which was confirmed by HRTEM image (Supplementary Fig. 2c). To evaluate the influence of oxygen-driven unzipping process on the electronic properties of nanobelts, we built a bottom-gated three-terminal device based on individual z-PNB of ~10 nm in thickness using the home-made Cu-grid mask method (Fig. 4a). As shown in Fig. 4b, when $V_g$ changing from 40 to $-40$ V, $I_{ds}$ increased up to over $10^{-8}$ A, corresponding to a turn-on state of z-PNB channel, which indicated a typical $p$-type semiconductor behavior. Lower gate leakage current ($I_{gate}$) of ~10 pA could make devices enable easier gate drive design and reduce power consumption[50]. A nonlinear relationship in output curves (inset of Fig. 4b) indicated that it is a typical Schottky contact of the device[4], which can be explained by the mismatch of work functions between z-PNBs (~3.8 eV, see Supplementary Fig. 9d) and contact metal Au (5.1 eV). Reported theoretical and experimental results showed that the ON/OFF ratio of PNR-based devices was up to ~1000[51,52]. However, our work showed that the z-PNBs devices presented an enhanced switching ratio behavior, reaching up to $4 \times 10^4$. The distinguished improvement behavior might be attributed to the bandgap expansion[13] (2.16 eV, see Supplementary Fig. 1g), compared with that of monolayer phosphorene (~2 eV)[4]. However, the hole mobility was scarified due to the Schottky barrier and large contact resistances caused by oxygen defects compared with mechanically exfoliated BP devices[4,10,24,53]. To some extent, unavoidable oxygen-induced defects in z-PNBs from electrochemical process will decrease the on-state current, leading to a lower ON/OFF ratio compared with that of $10^5$ achieved by mechanically exfoliated phosphorene nanosheet-based FETs[4,10,24]. Despite a lower hole mobility of 0.12 cm$^2$ V$^{-1}$ s$^{-1}$, such devices with a high ON/OFF ratio above $10^4$ and a low leakage current (~10 pA) properly met the requirements of active-matrix displays for potential high-performance military and avionics applications (see "Methods")[54]; while a high hole mobility with a maximum value over 1000 cm$^2$ V$^{-1}$ s$^{-1}$ was obtained at channel thickness of 109 nm of z-PNBs, obtained by a short exfoliation time (Fig. 4c), which was considered as a promising candidate for radio-frequency electronics[10]. The linear

relationship between $V_{ds}$ and $I_{ds}$ indicated that the thicker z-PNBs-based device showed a good Ohmic contact, leading to a higher hole mobility of $\sim1000\ cm^2\ V^{-1}\ s^{-1}$ (inset of Fig. 4c). Thickness-dependent behavior of a group of transfer curves of z-PNB devices was observed in Fig. 4d, indicating that it presents a trade-off relationship between hole mobility (red line) and ON/OFF ratio (blue line). It should be pointed out that the $I_{ds}$ increased as $V_g$ changed from negative to positive, when we used Al ($\Phi \sim 4.0$–$4.3\ eV$) as contact electrodes in the z-PNB devices. It indicates that the electrons, rather than holes, were the major carriers in the z-PNB devices. Thus, the z-PNB devices displayed n-type behavior with an electron mobility of $87\ cm^2\ V^{-1}\ s^{-1}$ (Supplementary Fig. 10a). This is due to the low work function of Al ($\Phi \sim 4.0$–$4.3\ eV$), which is well matched with that of z-PNBs ($\sim3.8\ eV$, see Supplementary Fig. 9d), and further decreases the Schottky barrier of electron transfer.

With the availability of both p-type and n-type, z-PNB devices showed great potential for complementary metal oxide semiconductors circuits[10,53]. It also demonstrated the stability of poly-methyl methacrylate (PMMA)-protected devices (inset of Fig. 4d), because both the ON/OFF ratio and hole mobility remained over 80% with the protection of PMMA even after 10 days. The electronic properties of oxygen-driven unzipping z-PNBs were analogous to other nanoribbon-based devices (Supplementary Fig. 10b).

In conclusion, our study has demonstrated the electrochemical unzipping method to produce z-PNBs with morphology control, opening doors in materials science and nanoelectronics. DFT calculation indicates that oxygen plays an important role during the formation of phosphorene nanosheets, nanobelts, and quantum dots. In a broader perspective where a variety of effects are predicted in z-PNBs, our simple, low-cost, and high productive approach with better control of oxidation may inspire a wide range and opportunity of research and applications in nanoelectronics.

## Methods

**Computational method**. All first-principles calculations are performed within the framework of spin-polarized DFT implemented in the VASP[55,56]. The exchange-correlation interactions are treated within the generalized gradient approximation of the PBE type[57]. Valence states of all atoms were expanded in a plane wave basis set with a cutoff energy of 400 eV[55]. A Gamma centered Monkhorst–Pack mesh of $4 \times 1 \times 1$ $k$ points was used for Brillouin Zone integration, where the AC direction is 4. Van der Waals interactions are considered by the DFT-D3 method for geometry optimization. Lattice geometries and atomic positions are fully relaxed until the forces on each atom are <0.01 eV/Å. Before unzipping the BP structure, a vacuum of $\sim17.8$ Å ($b = 20$ Å) was added in the direction ($b$ direction) normal to the monolayer to avoid spurious interactions between periodic replicas (Supplementary Fig. 6a). After unzipping BP structure, a vacuum of $\sim17.8$ Å ($b = 20$ Å) was added in the direction normal to the monolayer, with at least 12 Å vacuum in c direction after optimization separating the nanobelts to avoid interactions (Supplementary Fig. 6b).

**Preparation of z-PNBs**. Specifically, [BMIM]BF$_4$ (98%, HEOWNS), isopropanol (IPA, 99.7%, Beijing Chemical Works), N, N-Dimethylformamide (DMF, 99.5%, Beijing Chemical Works) were used directly without further purification. High-purity bulk BP (99.998%, $\sim1 \times 0.5$ cm) was purchased from the XF NANO, prepared by a high-pressure and high-temperature method, which is used as the precursor of our electrochemical exfoliation method. The bulk BP was inserted as anode into the ([BMIM]BF$_4$)/distilled water (without O$_2$ dissolved) solution, placed parallel to the Pt plate as counter-electrode with a separation of 2 cm. In our experiment, the ionic liquid was mixed with distilled water with weight ratio of 1:2. A careful experimental study reported that the exfoliated carbon nanoparticles could be controlled by changing the water/IL ratio[58]. In this study, we use different current densities to control the exfoliated products. Static current densities of $\sim0.1$ to $\sim0.5$ A cm$^{-2}$ were applied to the two electrodes using a DC power supply for 30 min. The products were sonicated under 100 W for 3 min, then centrifuged with distilled water (without O$_2$ dissolved), DMF (without O$_2$ dissolved) and IPA (without O$_2$ dissolved) for three times under 12000 r min$^{-1}$, respectively. Finally, the powders were carefully collected and dried under vacuum for the whole night. When the current density was <0.1 A cm$^{-2}$, the main product was nanosheets. When the current density range was $\sim0.2$-0.3 A cm$^{-2}$, the main products were

nanobelts. As the current density continued to increase to $\sim0.5$ A cm$^{-2}$, quantum dots were easily produced. The length and thickness of z-PNBs were also found to be influenced by the pristine bulk BP chunk and exfoliation time, respectively. Longer chunk could more possibly lead to longer z-PNBs (Supplementary Fig. 1j, k) and decreasing time of exfoliation led to thicker z-PNBs (Supplementary Fig. 1l).

**TEM and HRTEM**. For TEM, HRTEM, SAED, and EDS mapping tests, supernate of the z-PNBs after exfoliation was firstly dropped carefully on micro-grids, then they were dried under vacuum at 60 °C for 60 min, cooled down along with the oven subsequently. TEM images were obtained with H-7650B at 80 kV. HRTEM images, SAED patterns, and EDS mapping were acquired with JEM-2100F at 200 kV. Seventy-three TEM images of z-PNBs were collected for the size distribution (Supplementary Fig. 2f). For BP, the lattice parameters of $a = 3.314$ Å, $b = 10.473$ Å, and $c = 4.374$ Å[27–29]. Thus, the interplanar spacing of (002) and (200) could be calculated to be 0.2187 and 0.1657 nm, respectively. Then we could calibrate the (002) and (200) crystal plane in the SAED patterns (Fig. 1c and Supplementary Fig. 2a). The ratio between the (101) and (200) reflections was calculated using Gatan DigitalMicrograph (inset of Fig. 1c and Supplementary Fig. 2b). We performed Nano Measurer to measure the length and width of 73 z-PNBs (Fig. 1e and Supplementary Fig. 2f).

**XRD and XPS**. Followed by exfoliation, the z-PNBs dispersed in IPA (520 μL) were dropped onto the glass substrate in glove box and dried at 60 °C under vacuum for 1 h, then the sample was taken for XRD tests immediately. The powders of z-PNBs were collected after exfoliation and dried at 60 °C under vacuum for the whole night. XRD patterns were taken with Bruker D8 Advance XRD-7000 with Cu Kα radiation ($\lambda = 1.54178$ Å), scanning angle from 5 to 80°. XPS spectra were obtained with an ESCALAB 250Xi system using Al $K_\alpha$ as the excitation source. During the process of transferring samples for the XPS test, the z-PNBs are exposed to air for $\sim5$ min, leading to some additional oxidation in the XPS results. All binding energies were referenced to the C 1s peak at 284.8 eV. The fitting results of peak area of P 2p and O 1s were shown in Supplementary Tables 1 and 2, respectively. Different Lorentzian–Gaussian ratio—$\Sigma\chi^2$ of P 2p and O 1s was shown in Supplementary Table 3. The chi squared ($\chi^2$) quantity gives a measure of the goodness of fit between a set of experimental data points (Supplementary Table 3)[59]. Thus, XPS peak fitting was carried out using a mixed Lorentzian (10%)–Gaussian (90%) function after a Shirley background subtraction. The higher carbon and oxygen content in XPS survey was introduced by surfaces unavoidably carbon contamination[60].

**Raman, AFM, and optical analysis**. After exfoliation, the z-PNBs powder was dispersed in IPA. The solution was then dropped onto the 300-nm SiO$_2$/Si substrate, and then quickly transferred to glove box to make the solution absolutely volatilized. Then self-made instrument was used to carry the samples before Raman, AFM, and optical analysis. Raman spectra and polarized Raman spectra were collected with Horiba-Jobin-Yvon Raman system under 532-nm laser excitation with 1% power (0.13 mW). The ultrathin phosphorene nanosheet was hard to find the signal during the polarize Raman tests, thus we chose thick nanobelts as samples. Raman mapping images were performed with a step of 2 μm. AFM was performed using the Asylum Research Cypher AFM, AFM images of 56 z-PNBs were collected (Supplementary Fig. 3) The optical pictures were captured with Olympus BX 51M microscope.

**Mobility calculated needed for the three-terminal devices applied in active matrix**. From the equation[54]: $I_{ON}/I_{OFF} \geq 2V_{max}(t_F - t_L)/\Delta V_{px}(t_L)$, $t_L$ and $t_F$ represented the line section time and frame time, respectively. $V_{max}$ was the maximum voltage during the line selection time. $\Delta V_{px}$ was the voltage decay on the pixel capacitor during the hold time. $t_L$ was inversely proportional to the number of pixel rows[54], for a $64 \times 64$ matrix, $t_L \sim 0.3125$ ms[54], so when the matrix increased from $64 \times 64$ to $320 \times 240$, $t_L \sim 0.0833$ ms; $t_F$ was equal to the reciprocal of the corresponding frame rate under different resolutions, here frame rate was set to 50 Hz[54], thus, the $t_F \sim 0.02$ s. Besides, we took $V_{max}/\Delta V_{px} = 20$[54]. The mobility needed for active matrix was found to scale linearly with the number of rows of displays[54], for the resolution of $320 \times 240$, the mobility was at least 0.01 cm$^2$ V$^{-1}$ s$^{-1}$[54].

For different resolutions,

$320 \times 240$: $t_L \sim 0.0833$ ms, $t_F = 0.02$ s, $I_{ON}/I_{OFF} \geq 9.56 \times 10^3$, $\mu \geq 0.01$ cm$^2$ V$^{-1}$ s$^{-1}$
$640 \times 480$: $t_L \sim 0.0412$ ms, $t_F = 0.02$ s, $I_{ON}/I_{OFF} \geq 1.94 \times 10^4$, $\mu \geq 0.02$ cm$^2$ V$^{-1}$ s$^{-1}$
$1024 \times 768$: $t_L \sim 0.026$ ms, $t_F = 0.02$ s, $I_{ON}/I_{OFF} \geq 3.07 \times 10^4$, $\mu \geq 0.032$ cm$^2$ V$^{-1}$ s$^{-1}$

**UV tests**. The UV–vis absorption spectra of the z-PNB samples was measured by a Perkin Elmer Lambda 750 UV/Vis/NIR Spectrometer. A background of pure solvent (IPA) loaded in the same cell was subtracted from the spectra. The optical bandgap of z-PNBs could be approximately calculated by linear fitting the absorption data according to Tauc equation[61]: $ah\nu = A(h\nu - E_g)^{1/2}$, where $\alpha$, $h\nu$, $A$, and $E_g$ are the absorption coefficient, energy of incident light, constant and optical bandgap of z-PNBs, respectively.

**In situ ARPES and UPS**. We also used in situ angle-resolved photoemission spectroscopy (in situ ARPES) and ultraviolet photoelectron spectroscopy (UPS) to find the impact of oxygen content on the band structure. In situ ARPES and UPS characterizations were performed at the Photoelectron Spectroscopy Station in the Beijing Synchrotron Radiation Facility using a SCIENTA R4000 analyzer. A monochromatized He I light source (21.2 eV) was used for the band dispersion measurements. The experimental chamber's background vacuum was $3 \times 10^{-10}$ Torr, and all samples were kept at the temperature of 300 °C in the ultrahigh vacuum for 2 h to remove the carbon contamination on the sample surfaces from the atmosphere. For the in situ ARPES process, bulk BP chunk was chosen as the sample, and the oxygen was controlled at $10^{-3}$ Pa during the test. The in situ ARPES results (Supplementary Fig. 9a–c) revealed that the valence band decreased during the tiny oxidation. However, electronic structures of BP could be easily broken when exposed to oxygen for longer time (>30 min), so it was necessary to control the oxidation during the unzipping process. For UPS test, z-PNBs was firstly dispersed in IPA, then the solution was dropped onto the conductive substrate carefully in a glove box. The sample was kept by self-made instrument filled with Ar before the test. By applying a sample bias of −5 V, the sample work function was determined by the secondary electron cutoff at the low kinetic energy region. The work function of z-PNBs (3.8 eV) was lower than that reported by previous works due to partial oxidation (Supplementary Fig. 9d)[62].

**Device fabrication and measurement**. To study the electrical performance of z-PNBs produced by oxygen-driven unzipping method, a bottom-gated three-terminal device was fabricated, where individual z-PNB worked as conducting channel connected by source and drain electrodes. A highly p-doped Si (100) with resistivity <0.0015 Ω cm was chosen as bottom-gate electrodes covered by 90-nm-thick silicon oxide ($SiO_2$) as bottom-gate insulator. The dispersion containing z-PNBs was dropped on above-mentioned substrate in glove box. Then, 50-nm-thick Au source and drain electrodes were thermally evaporated through a copper grid shadow mask. This fabrication technique can effectively relieve the damage of oxidation on electrical performance during the device fabrication. All transfer and output curves were characterized with a probe station at room temperature using Aglient B1500 analyzer under $N_2$ atmospheres. For the air-stability study, we measured $I_{ds} - V_g$ curve of device based on z-PNB over a 10-day period with PMMA protected (inset of Fig. 4d), which was rotated onto the devices surface (3000 r min$^{-1}$, 40 s). The whole rotary coating process was conducted in glove box. The hole mobility of device is then calculated by the equation $\mu = [dI_{ds}/dV_g] \times [L/(WC_iV_{ds})]$, where $L$ is the channel length, $W$ is the channel width, where $C_i$ is the oxide capacitance of 3.84 F cm$^{-2}$, $V_{ds}$ is the drain to source bias, and $dI_{ds}/dV_g$ is the transconductance.

**Electrochemical test**. Chronopotentiometry curves were conducted on a CHI660E (Chenhua Co., Shanghai, China), using an IL/Water (weight ratio = 1:2) as the electrolyte. The current densities were set from 0.1 to 0.5 A cm$^{-2}$, the testing time was set at 60 s. The working and counter electrodes were a BP chunk and a platinum plate, respectively. The reference electrode was an Ag/Ag$^+$ electrode. The chronopotentiometry curves during the electrochemical process were shown in Supplementary Fig. 1h. As time goes on, the voltage increases gradually, indicating the resistance of BP is enlarged. Thus, the bandgap of BP is open during the whole exfoliation process.

## Data availability

All the data that supports the findings of this study are available from the corresponding authors upon reasonable request.

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

## Acknowledgements

This research is financially supported by National Key R&D Program of China (2016YFA0200200) and National Natural Science Foundation of China (Nos. 51672154). We thank Prof. Chun Li at Tsinghua University for the help on Raman experiments. We thank Prof. Han-shi Hu at Tsinghua University, Prof. Junyin Zhang, and Dr Shaoqiang Guo, Xiaofang Jia at Beihang University for discussions on theoretical calculations. We also acknowledge the Tsinghua Xuetang Talents Program for providing computational resources.

## Author contributions

H.C., D.X., and A.K.C. co-designed and supervised the research. Z.L. conducted the preparation and characterizations of phosphorene. Z.L. and W.L. conducted the DFT calculations. Z.L. and Y.S. conducted device fabrication, electrical properties measurements, and data analysis. J.W. did the ARPES and UPS measurements. All authors analyzed the data. H.C., Z.L., D.X., Y. S., and A.K.C. co-wrote the manuscript.

## Competing interests

The authors declare no competing interests.
