## [Peer Review File · Nature Communications]

Reviewers' comments:

Reviewer #1 (Remarks to the Author):

The manuscript describes a method for producing nanostructured black phosphorus via its electrochemical intercalation with BF₄⁻ ions followed by oxygen-driven unzipping. The authors claim their method is useful since it can relatively-easily produce nanoribbons of phosphorene (the main focus of the paper) and tuned to produce phosphorene nanodots and sheets. While there are interesting aspects of this work, as it is currently written there are a number of serious issues that must be addressed (or the claims modified) for it to be suitable for publication:

Major comments:

1. Most importantly, the authors claim of 'phosphorene nanoribbons' (in the title and throughout) is not fully justified for two reasons 1) lack of clear demonstration of 'nano'ribbons and 2) lack of clear demonstration of phosphorene. These are not pedantic comments: for the interesting properties and effects to emerge in phosphorene nanoribbons (as shown by the theory/simulation papers) their width must be a) narrow and b) must be uniform. Though it's hard to tell from the images, not many presented ribbons here could be considered to be much below 100 nm. Moreover many of the ribbons certainly have very non-uniform widths see many examples in ED2f and ED3 – or look more like fragments or scrolls (also how did you determine the width for the histograms if it varied?).

Also the ribbons, which have had more detailed experiments on are of considerably larger width e.g. Fig1a, Fig 1j, Fig 4. Also, notably, the AFM clearly shows the ribbons to be covered with fragments – what are these? Are they all like this? There are many apparent tip effects (or double stripes) in the AFM images, please discuss.

Also 'phosphorene' is a single layer of bP and there does not seem to be any demonstration of a single layer (i.e. truly phosphorene) ribbon produced by this method. Did you find any? On a connected note why does a ribbon with layer 2.4 nm correspond to 3 layers? if layer spacing is 0.5 nm?

The reason these comments are so important is that the authors claim their method is beneficial over that in the recent Nature paper by Watts etc. due to it being less complicated/time-consuming. This may be true but the method seems to produce ribbons of far less quality. Note in Watts the authors a) produce ribbons of far narrower widths (e.g. below 10 nm) b) do measurement that demonstrate the widths are uniform c) clearly demonstrate single layer ribbons d) their AFM images of ribbons have no fragments on them. In general their ribbons look much higher quality.

Please note I'm not saying the current paper is not useful – but the authors should either hone their method to produce much narrower, more uniform-width, higher quality ribbons, including single layers (i.e. to justify their claims and title) or discuss the difference between their ribbons and those in Watts explicitly in the paper. Note that electrochemical intercalation to produce larger width nanoribbons has already been reported before [<https://www.nature.com/articles/nnano.2015.194.pdf>] – this work should be discussed - however in this paper the authors more accurately describe their ribbons as 'nanobelts' seeing as the large widths imply the confinement and edge effects expected from ribbons would be absent.

2. If the electrochemical process produces intercalated bP, please include some data to show this happens e.g. XRD showing a layer increase, or maybe Raman spectra showing an associated change? This aspect of the mechanism is not clearly demonstrated. Please accordingly fix this sentence, also modifying wrt point 1 "After applying current, the BF₄⁻ was speeded to insert bulk BP and expand the interlayer distance to produce phosphorene, meanwhile, oxygen molecules

formed at the anode continuously unzipped the phosphorene into nanoribbons.”

3. It's interesting you can produce dots but more characterisation than Raman spectra is required please. bP has a very strong Raman response so any fragment of residual bP would be expected to dominate the spectra. Intuitively bP nanodots would be expected to show a large increase of the Raman peaks width, therefore further evidence is required to demonstrate that the dots shown in TEM are indeed comprised of crystalline phosphorene fragments.

4. When discussing the mechanism you should discuss this paper <https://iopscience.iop.org/article/10.1088/2053-1591/1/4/045041> seeing as their discussion of the mechanism is similar to yours and from 2014.

5. Given your proposed mechanism why do you not find any narrow ribbons?

6. Please provide some measurements of the air stability of the nanostructures that are produced as this determines their usefulness.

Minor comments

1. I don't agree with the statement in the first line of the abstract or paper. While phosphorene is interesting there are many 2D materials that precede its discovery and are post-graphene (notably TMDs) and arguably more useful.

2. Nanoribbons are not 1D they are 2D strips.

3. In some places though the discussion is nice, it is overlong and distracts from the results e.g. I'm don't think such a long description of the black phosphorus crystal structure is needed.

Reviewer #2 (Remarks to the Author):

The authors reported a top-down electrochemical strategy to produce one-dimensional zigzag phosphorene nanoribbons from the 3D bulk black phosphorus. The methods are reasonable and supported by theoretical DFT calculations. The results are interesting and the manuscript is well written. I recommend publishing after a minor revision after appropriately address the following.

1. The authors reported DFT calculations results in Fig. 3 and extended Fig. 6&7. However, in the DFT methodology session, essential information is missing. For example, the simulation cell size used in the calculations. The cell size should be big enough to avoid interaction between O₂ (or H₂O) molecules between periodic images resulted from periodic boundary condition.

2. The author mentioned the vacuum space is more than 10 Å. This needs to be more specific, especially for a slab model with adsorption molecules. 10 Å is at the very small end of vacuum range to be considered in a slab model with adsorption species, given that the layer-spacing in BP is 5.3 Å. The authors are recommend to discuss this and verify that the vacuum space in their calculations are sufficient.

3. Not many literature work has been reported for 1D phosphorene nanoribbon synthesis. Ref #2 as an early work reporting a top-down process in producing discrete phosphorene nanoribbons. The authors are encouraged to compare their methods with Ref. #2, such as advantages of each method.

Reviewer #3 (Remarks to the Author):

The paper deals with the fabrication of zigzag-phosphorene nanoribbons via electrochemical

oxidation. Due to importance of 2D materials and processed, size-modulated material in particular, this article is of great interest to a broad readership in natural science, materials science and physics. Especially anisotropic phosphorene nanoribbons are expected to be of interest due to predicted interesting and record-breaking properties. Whether these predictions are on any value can only be judged if such ribbons are available. In this article an elegant way is described to achieve such ribbons on a large scale. The explained mechanism is clever and straight forward. The characterization of the ribbons is performed using state-of-the-art techniques, like AFM, Raman Spectroscopy, TEM including electron diffraction which prove the existence, type and crystallinity of the products. Equation 1 is trivial and can be removed to my point of view. A very impressive and fundamental piece of this work are the quantum chemical calculations on all intermediates during oxidation and its interpretation. Especially the relation of the current density of the experiment and the final products are of immense value. This can directly be used for effective scale up and commercialization. Indeed, very impressive. The authors also showed first applications of their ribbons in FET devices which will gain interests in different communities. All experiments are done with high accuracy and data are interpreted with care including suitable statistics. To my point of view researchers can reproduce the given experiments based on the statements and descriptions given in the manuscript.

Upon reading this excellent manuscript only a few questions arose. Is the preparation mechanism of bulk black phosphorus critical or product-directing for the formation of phosphorene ribbons? There are several synthesis procedures like high pressure, complex annealing and tempering and a mineralizer based way (Sn/SnI₄ as precursor) to synthesize it. Due to the variations and the synthesis history during formation the original procedure might be of interest. I recommend to discuss the different procedures briefly in the beginning and to check the way how the black phosphorus used in this manuscript was synthesized. I believe that a different stress-strain and impurity level can directly influence the oxidation behavior in the final product.

One major point to be readjusted is the name of the products. Due to the fact that at the edges of each ribbon P-O bonds of different kinds (P=O; P-OH) are formed there is a certain oxygen content present in each ribbon. The thinner and smaller the ribbons are the higher the ratio oxygen vs. phosphorus will be. I suggest to call the resulting species therefore phosphorene oxide nanoribbons instead of phosphorene nanoribbons in the entire manuscript.

In sum, I recommend to publish this article in Nature Commun. in its present form, maybe addressing the questions I asked in the last part of my comments.

Response letter

Authors' response to all reviewers:

First of all, we would like to express our appreciation to all reviewers for their positive comments on our manuscript and also for their questions and suggestions to help us improve its quality. After carefully studying the comments from all reviewers, we have added more analysis and discussion to support our claims and enhance the scientific importance. Below is the point-to-point response to each reviewer's comments and concerns. Authors' response was colored in light blue and quote of addition/revision of manuscript was colored in purple.

Reviewer #1:

The manuscript describes a method for producing nanostructured black phosphorus via its electrochemical intercalation with BF₄⁻ ions followed by oxygen-driven unzipping. The authors claim their method is useful since it can relatively-easily produce nanoribbons of phosphorene (the main focus of the paper) and tuned to produce phosphorene nanodots and sheets. While there are interesting aspects of this work, as it is currently written there are a number of serious issues that must be addressed (or the claims modified) for it to be suitable for publication:

Major comments:

1. **1** Most importantly, the authors claim of 'phosphorene nanoribbons' (in the title and throughout) is not fully justified for two reasons 1) lack of clear demonstration of 'nano' ribbons and 2) lack of clear demonstration of phosphorene. These are not pedantic comments: for the interesting properties and effects to emerge in phosphorene nanoribbons (as shown by the theory/simulation papers) their width must be a) narrow and b) must be uniform. Though it's hard to tell from the images, not many presented ribbons here could be considered to be much below 100 nm. Moreover many of the ribbons certainly have very non-uniform widths see many examples in ED2f and ED3 – or look more like fragments or scrolls (also how did you determine the width for the histograms if it varied?).

Reply: Many thanks for the comment.

(i) Demonstration of 'nano' ribbons: Nanoribbons are usually thought to have narrow width. However, some papers have reported that the width of nanoribbons varies from tens to thousands of nanometers [R1-R5]. We selected some paper published on high IF journals, shown in Table R1 below. This indicates that the nanostructure of the aforementioned widths can be called nanoribbons in broad terms, which is well recognized by academic circles, including both authors and editors. Therefore, we called the as-synthesized products as nanoribbons, just following past definitions of nanoribbons. However, we agree with the Reviewer 1 and have changed nanoribbons into "nanobelts", just as you suggested in the followed comment-**2** (which is recommend in the paper [<https://www.nature.com/articles/nano.2015.194.pdf>])

Table R1 The description of "nanoribbons" reported recently.

Description of nanoribbons	References
Title: High-performance thin-film transistors using semiconductor nanowires and nanoribbons. "CdS nanoribbons were synthesized usingThe resulting materials are predominantly	[R1]-p278

nanoribbons having thicknesses of 30–150 nm, widths of 0.5–5 μm , and lengths of 10–200 μm (Supplementary Fig. S4).”	
Title: Vapor-liquid-solid growth of monolayer MoS_2 nanoribbons. “...yielding highly crystalline ribbons with a width of few tens to thousands of nanometres.”; “straight and kinked narrow ribbons with widths of a few tens of nanometers, and lengths ranging from a few to tens of micrometers are grown (see Supplementary Fig. 2i for the width distribution). The step height of a majority of these ribbons is ~ 0.8 nm,...”	R2-p536
Title: Aharonov-bohm interference in topological insulator nanoribbons. “the as-grown Bi_2S_3 nanoribbons with thickness of 25-100 nm and width ranging from 50 nm to several micrometres.”	R3-p226
Title: Far-field coherent thermal emission from polaritonic resonance in individual anisotropic nanoribbons. “...The thickness of the ribbons was fixed at 100 nm to suppress the broadband emission.” “The realized widths of the fabricated nanoribbons were 11.5 and 6.28 μm (Fig. 2).”	R4-p2,3
Title: Anisotropic in-plane thermal conductivity of black phosphorous nanoribbons at temperatures higher than 100 K. “Figure 3. Temperature-dependentThickness (t)/width (W) of ZZ and AC nanoribbons are 170/540 nm and 170/590 nm, respectively. .. 170 (t)/540 nm(W) and 310 (t)/540 nm (W)...170 (t)/590 nm (W) and 270 (t)/420 nm (W).”	R5-p4

(ii) Demonstration of phosphorene: Just as the reviewer mentioned, the phosphorene was theoretically considered as a monolayer of black phosphorus [R6]: “Curiously, this name does not reflect the chemical nature of the material, which has no sp^2 bonds; instead, it comes from its conceptual similarity to graphene — the IUPAC name should be 2D phosphorene.”[R6-p1] Also this paper called monolayer to 5 layer black phosphorous as phosphorene (shown in Figure 3 in [R6]-p7). In addition, another important paper also gave a broad definition of phosphorene, i.e., “In broad terms, few-layer (<10 layers) black phosphorus can be defined as phosphorene, much the same as graphene.” ([R7]-p982). Also the definition of phosphorene is provided as “Phosphorene, the single- or few-layer form of black phosphorus, was recently discovered as a two-dimensional (2D) layered material holding great promise for applications in electronics and optoelectronics.” [R8]-p1)

Therefore, we cited the detailed definition of phosphorene in the beginning of the text as: “Phosphorene, which is a mono-layer of black phosphorus (BP) in the strictest term, or few-layer (<10 layers) form of black phosphorus in a broad sense¹⁻³, has a natural band gap, unlike graphene, and has aroused great interest^{1,4-18}.”

(iii) The uniformity and width: The observed non-uniform widths are mainly due to the viewing angle problem caused by the leaning of the nanobelts on the edge of hole of Cu grid, and the ultrasonic treatments after electrochemical exfoliation. The statistic results from TEM images showed that $\sim 25\%$ of the z-PNRs widths are below 100 nm, $\sim 45\%$ below 500 nm, $\sim 74\%$ below 1 μm .

(iv) The fragments on covered on the ribbons in AFM images are caused by ultrasonic treatment after electrochemical exfoliation in the synthetic procedure. Similar phenomena have also been

observed in many other papers. The scrolls ribbons are mainly attributed to strain, resulting from the bending/twisting of the ribbons sitting on TEM grid. This phenomenon of fragments was also observed in the reference [R7] (This article is Ref.28 in our old version (Sun, J., et al. *Nature Nanotechnology* 2015, 10, 980.) in our manuscript, also recommended to be read as [https://www.nature.com/articles/nnano.2015.194.pdf]). From the AFM image of this article ([R7]-p982), we can observe irregular shaped phosphorene sheet with white fragments, shown as follows. Similar phenomenon of scrolls is “commonly observed in ribbon-like structures”, such as Fig. S4 of [R1]. We have also added a paragraph in the title of Supplementary Figure 1, as “Note: the scrolls are caused by strain from the bending/twisting of the ribbons on TEM grid, while the fragments are attributed to ultrasonic treatment after electrochemical exfoliation.” In fact, this phenomenon was also observed from the AFM images of other 2D van der Waals materials, such as graphene, synthesized in solution [R9-R13]. Corresponding illustration is shown in the title of Figure 4, as “The presence of irregularly shaped sheets is result of the sonication of the fused sheets, which fragments upon sonication.”

Figure 2 | Evidence of monolayer and few-layer phosphorene. a. AFM image of monolayer phosphorene. Scale bar, 2 μm . b. TEM image of trilayer phosphorene nanoribbon showing the long side. Scale bar, 200 nm. Inset: HRTEM image of the edge. c. HRTEM image of selected area of b with measured lattice spacing. d. Schematic of square phosphorene (top view) with labelled bond lengths. e. AFM image of monolayer and few-layer phosphorene. Scale bar, 2 μm . f. Measured thickness of the phosphorene in a. g. XRD pattern of black phosphorus and phosphorene. h. Amplification of XRD pattern of black phosphorus and phosphorene in the low-angle range (dashed red rectangle in g).

(Note: This AFM image is from Fig. 2e of [R7] (also cited as Ref. 28 in our old version; Ref.2 in revised version)

Figure 4 | AFM image of graphene in tapping mode. a. Topography image (600 \times 600 nm) (height scale, 0–10 \AA ; scale bar, 100 nm). b, c. Height profiles (600 nm along the x-axis) obtained from positions b and c indicated by white arrows in a. To the bottom of the image, a narrow ridge, of around 1×10^{-2} m width, extends beyond the bulk of the sheet. The profile across this point of the image is fully consistent with those observed elsewhere and clearly shows the narrow valley between the ridge and the neighboring sheet. The presence of irregularly shaped sheets is a result of the sonication of the fused sheets, which fragment upon sonication.

[R9] From the paper, Choucair, M., et al. Gram-scale production of graphene based on solvothermal synthesis and sonication. *Nature Nanotechnology* 4, 30-33 (2009).

[R10] Englert J. M., et al. Covalent bulk functionalization of graphene. *Nature Chemistry* 2011, **3**, 279 (2011).

[R11] Tung, V. C., et al. High-throughput solution processing of large-scale graphene. *Nature Nanotechnology* **4**, 25 (2009).

[R12] Zhao, T., et al. Ultrafast growth of nanocrystalline graphene films by quenching and grain-size-dependent strength and bandgap opening. *Nature Communications* **10**, 1-10 (2019).

[R13] Seo, D. H., et al. Anti-fouling graphene-based membranes for effective water desalination. *Nature Communications* **9**, 683 (2018).

This TEM image is from [R1]: Duan, X., et al. High-performance thin-film transistors using semiconductor nanowires and nanoribbons. *Nature*, **425**, 274-278 (2003): “**Fig. S4.** (a) Low-resolution TEM image of a CdS nanoribbon. Scale bar: 2 μm . A patterned contrast observed in the TEM image is due to strain resulting from the bending/twisting of the ribbons sitting on TEM grid, which is commonly observed in ribbon-like structures (Pan, Z., Dai, Z., Wang, Z., *Science*, **291**, 1947 (2001)). (b) High resolution TEM image shows that the nanoribbon has a single crystal structure almost free of defects and surface oxide layer. Scale bar: 5 nm. This high crystalline quality explains the exceptional performance of CdS nanoribbon TFT devices.”

(v) To sum the width and length of a series of nanoribbons, we used the software named Nano Measure to measure the length and width of the relatively flat part of the nanoribbons (see example: our Supplementary Fig. 2f- (34)). the observed non-uniform widths are mainly due to the viewing angle problem caused by the twist of nanosheets and the ultrasonic treatments after electrochemical exfoliation. The corresponding description has been added into the title of Supplementary Figure 1. “Note: the scrolls are caused by strain from the bending/twisting of the nanobelts on TEM grid, while the fragments are attributed to ultrasonic treatment after electrochemical exfoliation.”

Supplementary Figure 2f-(34). The measurement of phosphorene nanoribbon presents scroll feature.

② Also the ribbons, which have had more detailed experiments on are of considerably larger width e.g. Fig 1a, Fig 1j, Fig 4. Also, notably, the AFM clearly shows the ribbons to be covered with fragments – what are these? Are they all like this? There are many apparent tip effects (or double stripes) in the AFM images, please discuss.

Answer: Many thanks for the instructive comments.

(i) We did TEM (Fig. 1a) measurement under 80kV and corresponding HRTEM (Fig. 1b), electron diffraction (ED) (Fig. 1c) were measured under 200 kV, on large width phosphorene nanobelts, because narrow nanoribbons can be destroyed under both 80 kV and 200 kV

electron-beam irradiation within several minutes. The thinner and smaller ribbons can be more easily destroyed by electron-beams. This phenomena have also be observed in Ref.2-p2 of Methods- as “We note that degradation of ribbons after electron-beam exposure occurred at both 60 kV and 200 kV after several minutes.”

(ii) To avoid the high-level oxidation of phosphorene nanobelts, we carried out polarization-resolved Raman scattering spectroscopy measurement on wider nanobelts (Fig. 1j), because the laser of Raman Spectrometer irradiated on the sample for a long test time (about 24 min) which was caused by the angle of laser irradiation changing from 0° to 360° continuously with one time each scanning measurement per 15°.

(iii) The wider nanoribbons were found to be more suitable to fabricate three-terminal devices especially for the Cu-mask method (Fig. 4). Also we can notice other nanoribbons reported for FETs are also very large, such as, [R5]: “Figure 3. Temperature-dependentThickness (t)/width (W) of ZZ and AC nanoribbons are 170/540 nm and 170/590 nm, respectively. .. 170 (t)/540 nm(W) and 310 (t)/540 nm (W)...170 (t)/590 nm (W) and 270 (t)/420 nm (W).”

(iv) About the fragments: please refer to our response in above comment ①-(iv).

(v) About the tip effects: Because we used new AFM tips in the measurement, the broken tip might be excluded for the tip effect. However, just as discussed above, the fragments, shown in the AFM images, are generated by under sonication treatment. Thus, the tip effect observed in the AFM images might be attributed to the tip contaminations. [R14,R15]. This discussion is added in the revised version as “The observed tip effect observed in the AFM images might be caused by tip contaminations, not by broken tip, due to new AFM tip was used in the measurements.”.

② Also ‘phosphorene’ is a single layer of bP and there does not seem to be any demonstration of a single layer (i.e. truly phosphorene) ribbon produced by this method. Did you find any? On a connected note why does a ribbon with layer 2.4 nm correspond to 3 layers? if layer spacing is 0.5 nm?

Answer: Many thanks for this insightful comment.

(i) About the monolayer phosphorene: Although there are a few single-layer black phosphorene nanobelts, we can indeed observe single-layer black phosphorene in AFM images (Please check it in Supplementary FFig. 3-20 in old version). We also provided this AFM image as inset of Fig. 1f in the revised version. The statistical results from AFM images showed that thickness of ~90% of the z-PNRs are below 5 layers, ~60% below trilayer layers and ~5% for monolayer. It should be pointed out that the phosphorene nanoribbons reported by Watts, et al. (Ref.2-p218 in our old version, Nature 2019, 568, 216) are not 100% monolayer phosphorene, but “most (around 65%) of the PNRs are monolayer; 84% are trilayer or fewer.” Even though there are still multi-layer nanoribbons, most of the nanoribbons reported by Watts et al., are monolayer nanoribbons, which is indeed better than ours.

Supplementary Figure 1f Thickness distribution diagram from the AFM images of 56 z-PNBs. Inset: AFM image of a typical ribbon onto a 300 nm SiO₂/Si substrate with the thickness of ~2.4 nm (left) and ~0.8 nm (right), corresponding to trilayer and monolayer phosphorene, respectively.

(ii) About the definition of monolayer thickness of 0.84 nm [R7] (This is cited from Ref. 28 in old version, or Ref. 2 in revised version). According to the previous report [R7], we defined the thickness of monolayer phosphorene as ~0.84 nm. This definition is widely used [R7,R16-R20]. The theoretical layer spacing of phosphorene was 0.53 nm, however, “roughness and sample–tip interaction difference between SiO₂ and black phosphorus can lead to a significant overestimate of thickness” [R20]. Thus the thickness for monolayer phosphorene was experimentally more than 0.53 nm.

Table R2. Thickness of Monolayer phosphorene reported recently.

Thickness of Monolayer phosphorene	Reference
0.84 nm	R7
0.85 nm	R16
0.85 nm	R17
0.85 nm	R18
0.85 nm	R19
1.5 nm	R20

The monolayer thickness of other 2D materials, such as graphene and MoS₂, were also experimentally considered to be higher than that theoretical results [R21-R26].

Table R3. Theoretical and experimental results of the thickness of other monolayer 2D materials.

Other materials	2D	Theoretical thickness of monolayer	Experimental results of monolayer	Reference
Graphene		0.34 nm (R21)	0.5 nm	R22
Graphene		0.34 nm	0.6 nm	R23
MoS ₂		0.65 nm (R24)	0.72 nm	R25
MoS ₂		0.65 nm	0.8 nm	R26

④ The reason these comments are so important is that the authors claim their method is beneficial over that in the recent Nature paper by Watts etc. due to it being less complicated/time-consuming. This may be true but the method seems to produce ribbons of far less quality. Note in Watts the authors a) produce ribbons of far narrower widths (e.g. below 10 nm) b) do measurement that demonstrate the widths are uniform c) clearly demonstrate single layer ribbons d) their AFM images of ribbons have no fragments on them. In general their ribbons look much higher quality.

Please note I'm not saying the current paper is not useful – but the authors should either hone their method to produce much narrower, more uniform-width, higher quality ribbons, including single layers (i.e. to justify their claims and title) or discuss the difference between their ribbons and those in Watts explicitly in the paper. Note that electrochemical intercalation to produce larger width nanoribbons has already been reported before [<https://www.nature.com/articles/nnano.2015.194.pdf>] – this work should be discussed - however in this paper the authors more accurately describe their ribbons as ‘nanobelts’ seeing as the large widths imply the confinement and edge effects expected from ribbons would be absent.

Answer: Many thanks for the comment. We have compared the two methods as follows:

(i) It is true that the work reported by Watts. et. al demonstrated a Li-ion intercalation method to produce much narrower, more width uniform, higher quality ribbons, including single layerd phosphorene nanoribbons, just as you listed: a) produce ribbons of far narrower widths (e.g. below 10 nm), b) do measurement that demonstrate the widths are uniform, c) clearly demonstrate single layer ribbons, d) their AFM images of ribbons have no fragments on them. However, at the same time, their synthesis has tedious and rigorous steps, followed a two-step process: a) First, intercalation step: BP crystals are intercalated with lithium ions via a low temperature, ammonia-based method, cooled to -50°C and high-purity ammonia (NH₃) was condensed onto the BP and lithium to form a Li:NH₃ solution. The BP was left submerged in the Li:NH₃ solution for about 24 h, allowing intercalation. b) Second, ultrasonic treatment for 1 h and centrifuged at low acceleration. It should be pointed out that before the intercalation step, BP crystals were first outgassed at 100 °C under dynamic vacuum (less than 10⁻⁶ mbar) for over week.

In contrast, our method has some advantages, such as, a) our work shows a simple and **timeless** method to prepare zigzag-phosphorene nanoribbons at room-temperature, which “can directly be used for effective scale up and commercialization” (this evaluation is from the third reviewer). b) We firstly propose this electrochemical method via controlling the current density to prepare phosphorene nanoribbons, accompanied with an unzipped mechanism, supported by detailed theoretical calculations. c) The phosphorene nanoribbons find applications in FET devices. d) Besides nanoribbons, we can also obtain quantum dots and nanosheets via changing the current density.

In order to limit the length of text, we have also added a brief comparison in the text as follows: “The reported ionic scissoring method⁵ can generate high-quality, individual PNR; most (around 65%) are monolayer with PO_x content of 15 at.%, but the synthesized steps are time-consuming and rigorous. By contrast, our electrochemical method has simple steps; and most (around 60%) of belts are less than trilayer and lower PO_x content (~9.9 at.%).”

(ii) The suggested paper [<https://www.nature.com/articles/nnano.2015.194.pdf>] was cited in our manuscript as [R7] (also Ref.28 in our old version, or Ref.2 in revised version). This work reports

the formation of monolayer and few-layer phosphorene with very irregular shapes (Fig. 2a, 2e in [R7]), because it was synthesized by only sonication in a sonic bath (Branson 5210 Ultrasonic) for 10 h (Methods). Obviously, this method is not an electrochemical method, but a sonication method. This leads to very irregular shaped products, because of breaking P-P via a sonication method in random directions.

However, our electrochemical method leads to zigzag-phosphorene nanoribbons with high crystallinity and low oxidation level.

Indeed, the observed phosphorene nanoribbons mostly have relatively larger width and lower uniformity than those of being reported by Watts. et. Al. (Ref. 2 in our old version, or Ref. 5 in revised version). The statistical results from TEM images showed that ~25% of the z-PNRs in width are below 100 nm, ~45% below 500 nm, ~74% below 1 μm (Fig. 1e and Supplementary Fig. 2f), and AFM images showed that ~90% of the z-PNRs in thickness are below 5 layers, ~60% below tri-layers, and ~5% for monolayer (Fig. 1f and Supplementary Fig. 3). The monolayer phosphorene and narrow nanoribbons (<100 nm) could be produced by our electrochemical method. Also we find applications in FET devices from our zigzag-phosphorene nanoribbons.

In order to limit the length in text, we also add a brief comparison in the text as follows: “Cui et al² reported a liquid-phase exfoliation method to form phosphorene by immersing the BP into a sonic bath, followed by ultrasonically exfoliated. This method can form irregular shaped few-layers phosphorene, because breaking P-P is random by a sonication method. However, as long as the current density is changed, our electrochemical method can generate not only z-PNBs, but also quantum dots and nanosheets.”

2. If the electrochemical process produces intercalated bP, please include some data to show this happens e.g. XRD showing a layer increase, or maybe Raman spectra showing an associated change? This aspect of the mechanism is not clearly demonstrated. Please accordingly fix this sentence, also modifying wrt point 1 “After applying current, the BF₄⁻ was speeded to insert bulk BP and expand the interlayer distance to produce phosphorene, meanwhile, oxygen molecules formed at the anode continuously unzipped the phosphorene into nanoribbons.”

Answer: Many thanks for your suggestion to measure the XRD or Raman of intercalated BP to understand the electrochemical process. However, our University is still close at present time, not even knowing when it will be open, due to Covid-19 epidemic. However, we can provide other evidence that the BF⁻ intercalation mechanism does occur, as follows:

(i) “The BF₄⁻ intercalation process can be manifested by the chronopotentiometry curves during the exfoliation process at different current densities ranging from ~0.1 to 0.5 A cm⁻² (Supplementary Fig. 1h). It was found that the voltage first drops rapidly, and then rises slowly. The decrease in voltage is due to the increase in surface area of black phosphorus, caused by BF₄⁻ intercalation into the BP crystal, while the increase in voltage is due to the increase in the bandgap of phosphorene compared with that of BP crystal⁴, caused by the intercalation of ions^{2,42}.” The predicted direct bandgap of phosphorene is ~2 eV, while the bandgap of BP crystal is ~0.3 eV [R27]. This above discussion is added in the text.

Supplementary Figure 1h Chronopotentiometry curves during the exfoliation process at different current densities from ~ 0.1 to 0.5 A cm^{-2} . The resistance increased with longer exfoliation time, leading to the larger bandgap. It is noted that the origin sharp decrease is due to the increased surface area. Inset: The enlarged chronopotentiometry curve of $\sim 0.1 \text{ A cm}^{-2}$.

(ii) Some branch-like cracks along one direction could be observed from our TEM image, which were generated only by an electrochemical exfoliation but without sonication treatment (Supplementary Fig. 2d). We have also added this sentence in the text, as follows, “Furthermore, the intercalation intermediate step of the electrochemical synthesis method without ultrasonic treatment can be directly observed by the TEM observation (Supplementary Fig.2d).”

Supplementary Figure 2d. TEM image of a z-PNR electrochemical exfoliation without sonication after.

(iii) It is worth mentioning that the XRD of intercalation compounds is likely not to reflect the change [R28,R29], the views can also be found from below literature:

“However, after intercalation, while the peaks characteristic of pristine BP have almost disappeared, as expected, no new peaks corresponding to an expanded crystal lattice are visible. We attribute this to the fact that intercalation is limited to $\sim 10 \mu\text{m}$ thick layers near the surface, while X rays probe the entire volume of the crystals.” ([R28]-p3)

“Furthermore, successful intercalation of black phosphorus manifested only as a subtle change in the associated X-ray diffraction (XRD) pattern. Another recent black phosphorus intercalation study, using the same ammonia-based intercalation method that we used here, again found no well defined layer increases typical of layered materials upon intercalation.” ([R29]-Methods section _Production of phosphorene nanoribbons. This paper is cited as Ref.2 in old version, or Ref. 5.)

(iv) It should be noted that black phosphorus is only oxidized and degraded under the ambient conditions, but not to generate nanoribbons, quantum dots, and nanosheets. The intercalation of BF_4^- ions into layers of BP crystals before O_2 unzipping is important to form nanobelts. Thus, we believe that the intercalation of BF_4^- will expand the layer-spacing of BP crystal and then O_2 could unzip the nanosheets to form different nanostructures varied from nanosheets to quantum dots.

(v) Also we changed the sentence of “After applying current, the BF_4^- was speeded to insert bulk BP and expand the interlayer distance to produce phosphorene, meanwhile, oxygen molecules formed at the anode continuously unzipped the phosphorene into nanoribbons.” into “When the power is turned on, the BF_4^- anions move toward the anode (BP) and are inserted between BP layers, which results in the expansion of the layer spacing of the BP crystals; and the more oxygen formed at anode due to electrolysis of water will accelerate the unzipping P-P bonds of BP.”

3. It’s interesting you can produce dots but more characterisation than Raman spectra is required please. bP has a very strong Raman response so any fragment of residual bP would be expected to dominate the spectra. Intuitively bP nanodots would be expected to show a large increase of the Raman peaks width, therefore further evidence is required to demonstrate that the dots shown in TEM are indeed compromised of crystalline phosphorene fragments.

Answer: Many thanks for your suggestion to provide further evidence of dots. We wish we could do HRTEM to further characterize the crystallinity of quantum dots. As you may understand our University is still closed at present time, not even knowing when it will start normally, due to Covid-19 epidemic.

However, we can still use Raman to characterize the quantum dots to identify the layer number. We compared the Raman spectra of phosphorene quantum dots (PQDs) and reported PQDs and the as-synthesized phosphorene nanobelts. In the spectrum of PQDs, four Raman peaks at 364.0 (A_g^1), 440.3 (B_{2g}), 468.5 (A_g^2), and 521.0 cm^{-1} can be observed. The major scattering peak at 521.0 cm^{-1} is from the Raman peak of the silicon substrate. These Raman shifts of PQDs are larger than those reported at 361.1, 438.1, and 465.4 cm^{-1} in the literature [R30]. It is known that the A_g^2 mode shift to higher frequency is attributed to the thickness decrease [R31]. The frequency of A_g^2 mode of PQDs is slightly higher than the as-synthesized triple-layer zigzag-phosphorene nanoribbons, indicating the PQDs is a triple layer structure. Also the intensity ratio of A_g^1 and Si peak is ~ 0.2 , suggesting they are trilayered phosphorene [R32]. Thus, from Raman peaks position of A_g^2 mode and the intensity ratio of A_g^1 and Si, we can identify the quantum dots are trilayered BP structure.

Supplementary Figure 1f An example Raman spectrum of phosphorene quantum dots (PQDs) compared to z-PNRs. The A_g^2 peak at 468.5 cm^{-1} of phosphorene quantum dots, is very close the peak at 468.3 cm^{-1} of triple-layer phosphorene nanobelts. Also the intensity ratio PQDs of A_g^1 and Si peak is ~ 0.2 , suggesting it is also trilayer phosphorene with thickness of $\sim 2\text{ nm}$ of three layers.

4. When discussing the mechanism you should discuss this paper <https://iopscience.iop.org/article/10.1088/2053-1591/1/4/045041> seeing as their discussion of the mechanism is similar to yours and from 2014.

Answer: Many thanks for the comment. We have carefully read the suggested paper (<https://iopscience.iop.org/article/10.1088/2053-1591/1/4/045041>) [R33]. The authors report “the effect of surface passivation on phosphorene was explored using first-principle density functional theory (DFT) calculations. ...With H (or F, Cl, OH radicals) passivation on the phosphorus atoms, H and P form a strong σ bond and break the P-P bonds between the upper and lower half-layers of phosphorene.”([R33]-p2). They also demonstrated “that O (S or Se) does not break the P-P bonds where the variation of the P-P bond length is within $\pm 2\%$ compared to its original value.” ([R33]-p5).

However, our work reports the electrochemical method to unzip black phosphorus (BP) crystal via BF_4^- intercalation into the layers of BP, followed by O_2 unzipping mechanism, in which “the interaction between defect oxygen and water molecules can lead to P-P breaking, but oxidation alone does not break down the P-P bonds of phosphorene sheets.” Obviously, our work is quite different from above mentioned paper in mechanism. This reported work [R33] supports our DFT results in some way. In our theoretical results, O_2 cannot unzip the phosphorene independently. The unzipping process can only be completed with the participation of H_2O . This reported work does not show the O_2 adsorption process, along with the continued O_2 dissociation and P-OH formation. The energy configuration mentioned is incomplete, so it can not explain which configuration is most stable. However, we have provided complete energy configurations, from O_2 adsorption to P-OH formation, to discuss the stability, finally to zigzag phosphorene nanoribbons. Our theoretical calculations are fully consistent with experiments.

We also add the above discussion in the revised text, as follows, “A previous DFT study indicates that H, OH, F, and Cl can act as scissors to cut phosphorene into nanoribbons or nanochains, while O, S, and Se atoms cannot⁴⁹. In contrast, our unzipping mechanism shows that oxygen unzipping phosphorene is only available under the condition of the formation of hydrogen bonds with H_2O . This can explain the electrochemical exfoliation process in air with H_2O .”

5. Given your proposed mechanism why do you not find any narrow ribbons?

Answer: Many thanks for your comment.

(i) First, we can experimentally produce relatively narrow nanoribbons down to 30 nm. Please check the supplied TEM image below.

Supplementary Figure 1f and Figure 2f-(11). Narrow phosphorene nanoribbon with a width of 30 nm.

(ii) Two factors influence the synthesis, namely, the quality of the exfoliated BP crystal and the current control in the electrochemistry method. The more accurate control of current density is achieved, the more accurate control of oxygen concentration can be realized. The O_2 concentration will affect the oxygen adsorption on the surface of black phosphorus, thereby controlling the distance of adsorption sites. Of course, just as the Reviewer 3 suggested that the BP crystal prepared methods will have different stress-strain and impurity defect, which will affect the physical adsorption sites of O_2 in the electrochemical process.

A large number of narrower nanobelts were not obtained, due to low oxygen concentrations of synthesis conditions. Low oxygen concentration is not enough to allow oxygen to be fully adsorbed on the surface **closer together**, and therefore, which cannot lead to unzip P-P bonds along zigzag direction in closer P-P sites. Thus, it leads to wider nanobelts. However, if the O_2 concentration is large enough, a larger current density is required, which may lead to rapid intercalation, and nanosheets are detached from the BP crystal to form quantum dots. By the way, the high O_2 concentration will lead to high oxygen-containing in the products. The experiment result of our “O/P ratio of 0.11 (Fig. 1h) which was much lower than the limit of 0.89 for preventing degradation” (line 128 in old version) also support the above conjecture. Thus, it is required to obtain narrow nanoribbons by using an instrument capable of accurately controlling current.

6. Please provide some measurements of the air stability of the nanostructures that are produced as this determines their usefulness.

Answer: Many thanks for your comment. Our University is still closed at present time, not even knowing when it will start normally, due to Covid-19 epidemic.

However, we know that phosphorene is easily to be oxidized from our electronic structure measurement (Supplementary Fig. 9). The measurement of BP ([R29] (i.e. Ref.2 in our old version, or Ref. 5 in the revised version) pointed out that “it should be noted that few-layer/monolayer phosphorene is less air stable than multilayer/bulk black phosphorus.” Abate et al. reported that “In fact, pristine BP is hydrophobic and it is only after the oxidation process to form POx that the surface becomes hydrophilic. This degradation is also highly thickness dependent. Single-layer BP (phosphorene) is the fastest to degrade (within minutes after exfoliation).” [R34].

Based on the above knowledge, we collected the products in Schlenk Flask under inert gas protection after the synthesis of phosphene nanobelts, shown as follows. This prevents oxygen

from damaging the sample, and makes it easy to carry the samples to conduct other measurement. Also the three-terminal devices were also conducted inside a glove box and coated with PMMA to make the device stable.

Figure. 1 Collection of products in Schlenk Flask under inert gas protection, due to the phosphorene is easily to be oxidized in air.

Minor comments

1. I don't agree with the statement in the first line of the abstract or paper. While phosphorene is interesting there are many 2D materials that precede its discovery and are post-graphene (notably TMDs) and arguably more useful.

Answer: Many thanks for the comment. This view of “post-graphene age” in the first sentence is from Nature News by E. S. Reich’s “Phosphorene excites materials scientists” (Nature 2014, 506, 19) [R35], which was expressed as “... dubs the post-graphene age, in which researchers are exploring alternatives in the hope of overcoming graphene's deficiencies.” However, in view of many 2D materials are prior to phosphorene discovery and are post-graphene, we cancel it in the first sentence in the revised version to avoid ambiguity.

2. Nanoribbons are not 1D they are 2D strips.

Answer: Thank you for your comment.

There are different views for the definition of 1D nanostructures, such as, (i) Guozhong Cao, *Nanostructures & Nanomaterials, Synthesis, Properties & Applications*, Imperial College Press, 2007; p110: “Chapter 4 One-Dimensional Nanostructures: Nanowires and Nanorods: ...In addition, one-dimensional structures with diameters ranging from several nanometers to several hundred microns were referred to as whiskers and fibers in the early literature, whereas nanowires and nanorods with diameters not exceeding a few hundred nanometers are used predominantly in the recently literature.” [R36]. (ii) [R29]–“Production of phosphorene nanoribbons”, (i.e., Ref.2 in old version, or Ref. 5 in the revised version, Nature 2019, 568, 216. Line 3: “Nanoribbons, meanwhile, combine the flexibility and unidirectional properties of one-dimensional nanomaterials, the high surface area of 2D nanomaterials and the electron-confinement and edge effects of both”. From the sentence, we know that the nanoribbons have the properties of both 1 D and 2 D. (iii) Besides, we also list some papers which called nanoribbons as one-dimensional materials or quasi-one-dimensional materials, shown as follow Table. In broad term, based on above knowledge, quasi-one-dimensional nanoribbons may be

appropriate. However, in strict definition of one dimension, we have removed the term of “1D” in the manuscript.

Table R4 Nanoribbons were regarded as 1D nanostructure

Description of nanoribbons about dimensional	Reference
One-dimensional	R37
One-dimensional	R38
Quasi-one-dimensional nanoribbons	R3
Quasi-one-dimensional	R39
Quasi one-dimensional structures	R40

3. In some places though the discussion is nice, it is overlong and distracts from the results e.g. I'm don't think such a long description of the black phosphorus crystal structure is needed.

Answer: Many thanks for the comments. According to your suggestions, we have shorten our manuscript in the revised version. However, we have had to add some discussions in the revised version based on all of the reviewers' comments, which are marked purple in the revised version.

Reviewer #2:

The authors reported a top-down electrochemical strategy to produce one-dimensional zigzag phosphorene nanoribbons from the 3D bulk black phosphorus. The methods are reasonable and supported by theoretical DFT calculations. The results are interesting and the manuscript is well written. I recommend publishing after a minor revision after appropriately address the following.

1. The authors reported DFT calculations results in Fig. 3 and extended Fig. 6&7. However, in the DFT methodology session, essential information is missing. For example, the simulation cell size used in the calculations. The cell size should be big enough to avoid interaction between O₂ (or H₂O) molecules between periodic images resulted from periodic boundary condition.

Answer: Many thanks for your recommendation of our work.

(i) Based on your suggestion, we add the essential information for DFT methodology section in the revised version, also shown as follows, and Supplementary Figure 6:

“Computational Method. All first-principles calculations are performed within the framework of spin-polarized density functional theory (DFT) implemented in the Vienna ab initio Simulation Package (VASP)^{55,56}. The exchange-correlation interactions are treated within the generalized gradient approximation of the Perdew-Burke-Ernzerhof (PBE) type⁵⁷. Valence states of all atoms were expanded in a plane wave basis set with a cut-off energy of 400 eV⁵⁸. A Gamma centered Monkhorst-Pack mesh of $4 \times 1 \times 1$ k points was used for Brillouin Zone integration, where the armchair direction is 4. Van der Waals interactions are considered by the DFT-D3 method for geometry optimization. Lattice geometries and atomic positions are fully relaxed until the forces on each atom are less than 0.01 eV/Å. Before unzipping the BP structure, a vacuum of ~ 17.8 Å ($b = 20$ Å) was added in the direction (b axis) normal to the monolayer to avoid spurious interactions between periodic replicas (Supplementary Fig. 6a). After unzipping BP structure, a vacuum of ~ 17.8 Å ($b = 20$ Å) was added in the direction (b axis) normal to the monolayer, with at least 12 Å

vacuum in another direction (c axis) after optimization separating the nanobelts to avoid interactions (Supplementary Fig. 6b).”

Supplementary Figure 6. Supercell size of typical configurations. **a** Before unzipping BP structure, a vacuum of $\sim 17.8 \text{ \AA}$ ($b = 20 \text{ \AA}$) was added in the direction (b axis) normal to the monolayer to avoid spurious interactions between periodic replicas periodic replicas, which could minimize images interactions for the all configurations. **b** After unzipping BP structure, a vacuum of $\sim 17.8 \text{ \AA}$ ($b = 20 \text{ \AA}$) was added in the direction (b axis) normal to the monolayer, with another direction (c axis) at least 12 \AA vacuum after optimization separating the nanobelts to avoid interactions. The black, red, and pink spheres represent phosphorus, oxygen, and hydrogen atoms, respectively. The dotted lines represent hydrogen bond.

Figure 2. The spacing is 5.3 \AA . The single layer thickness is $\sim 2.2 \text{ \AA}$. Thus, a vacuum of $20 - 2.2 = 17.8 \text{ \AA}$ was added.

(ii) The layer-spacing of phosphorus is 5.3 \AA . Thus, the vacuum of 17.8 \AA is 3.35 times of 5.3 \AA of the layer-spacing of BP. Thus, this vacuum of 17.8 \AA set lets the cell parameter can relax at the horizontal directions, before the unzipping P-P.

After unzipping, a vacuum of $\sim 17.8 \text{ \AA}$ ($b = 20 \text{ \AA}$) was added in the direction (b axis) normal to the monolayer. Another direction (c axis) at least 12 \AA vacuum separating the nanoribbons was added, to avoid interactions. Based on the unzipping mechanism via the oxygen chemisorption and the interaction between defect oxygen and water molecules to break P-P bonds, we have considered it in the calculation, then we added at least 12 \AA vacuum separating the nanoribbons. Please see the insets of Supplementary Figure 8. The vacuum of 12 \AA is 5.3 times of the P-P bond length of 2.244 \AA (another kind of P-P bond length is 2.224 \AA), which is longer enough to ensure the P-P is broken.

Also we list the vacuum space sets in recent calculation work on phosphorene, shown in Table R4 [R41-R50].

Table R4: List of the vacuum space reported recently.

Vacuum space (Å)	Reference
10	R41
10	R42
10	R43
10	R44
12	R45
12	R46
12	R47
12.9	R48
15	R49
20	R50

2. The author mentioned the vacuum space is more than 10 Å. This needs to be more specific, especially for a slab model with adsorption molecules. 10 Å is at the very small end of vacuum range to be considered in a slab model with adsorption species, given that the layer-spacing in BP is 5.3 Å. The authors are recommend to discuss this and verify that the vacuum space in their calculations are sufficient.

Answer: Many thanks for the comment. Detailed information is shown in above first response. We fixed the cell parameter in the direction normal to the monolayer phosphorene of 20 Å, which lead to at least ~17.8 Å vacuum. Considering the layer-spacing of phosphorene is theoretically 5.3 Å and previous reports [R41-R50], ~17.8 Å is 3.35 times of 5.3 Å of the layer-spacing of BP. Thus, this vacuum of 17.8 Å is large enough to eliminate the interaction between periodic images. The vacuum of another at least 12 Å was added to separate the nanoribbons, besides considering the chemisorption of O₂ and hydrogen-bonding of H₂O, which is 5.3 times of the P-P bond length of 2.244 Å (another kind of P-P bond length is 2.224 Å). This set is longer enough to ensure the P-P is broken. Also we list the vacuum space sets in recent calculation work on phosphorene, shown in Table R4 [R41-R50].

3. Not many literature work has been reported for 1D phosphorene nanoribbon synthesis. Ref #2 as an early work reporting a top-down process in producing discrete phosphorene nanoribbons. The authors are encouraged to compare their methods with Ref. #2, such as advantages of each method.

Answer: Many thanks for the suggestion. We compare the two methods as follows:

It was true that the work reported by Watts. et. al demonstrated a Li-ion intercalation method to produce much narrower, more uniform-width, higher quality ribbons, including single layers phosphorene nanoribbons, just as you list: a) produce ribbons with far narrower widths (e.g. below 10 nm), b) do measurement that demonstrate the widths are uniform, c) clearly demonstrate single layer ribbons, d) their AFM images of ribbons have no fragments on them. However, at the same time, the synthesis method has **tedious and rigorous steps**, followed a two-step process: a) First, intercalation step: BP crystals are intercalated with lithium ions via a low temperature, ammonia-based method, cooled to -50°C and high-purity ammonia (NH₃) was condensed onto the

BP and lithium to form a Li:NH₃ solution. The BP was left submerged in the Li:NH₃ solution for about 24 h, allowing intercalation. b) Second, ultrasonic treatment for 1 h and centrifuged at low acceleration. It should be pointed out that before intercalation step, BP crystals were first outgassed at 100 °C under dynamic vacuum (less than 10⁻⁶ mbar) for over week.

As a contrast, our method also own some advantages, such as, a) our work shows a simple and timeless method to prepare zigzag-phosphorene nanoribbons at room-temperature, which “can directly be used for effective scale up and commercialization” (this evaluation is from third reviewer). b) We firstly propose this electrochemical method via controlling the current density to prepare phosphorene nanoribbons, accompanied with an unzipped mechanism, supported by detailed theoretical calculations. c) The phosphorene nanoribbons find applications in FET devices. d) Besides nanoribbons, we can also obtain quantum dots and nanosheets via changing the current density.

In order to limit the length in text, we also add a brief comparison in the text as follows: “The reported ionic scissoring method⁵ can generate high-quality, individual PNR; most (around 65%) are monolayer with PO_x content of 15 at.%, but the synthesized steps are time-consuming and rigorous. By contrast, our electrochemical method has simple steps; and most (around 60%) of belts are less than trilayer and lower PO_x content (~9.9 at.%).”

Reviewer #3:

● The paper deals with the fabrication of zigzag-phosphorene nanoribbons via electrochemical oxidation. Due to importance of 2D materials and processed, size-modulated material in particular, this article is of great interest to a broad readership in natural science, materials science and physics. Especially anisotropic phosphorene nanoribbons are expected to be of interest due to predicted interesting and record-breaking properties. Whether these predictions are on any value can only be judged if such ribbons are available. In this article an elegant way is described to achieve such ribbons on a large scale. The explained mechanism is clever and straight forward.

The characterization of the ribbons is performed using state-of-the-art techniques, like AFM, Raman Spectroscopy, TEM including electron diffraction which prove the existence, type and crystallinity of the products. Equation 1 is trivial and can be removed to my point of view. A very impressive and fundamental piece of this work are the quantum chemical calculations on all intermediates during oxidation and its interpretation. Especially the relation of the current density of the experiment and the final products are of immense value. This can directly be used for effective scale up and commercialization. Indeed, very impressive. The authors also showed first applications of their ribbons in FET devices which will gain interests in different communities. All experiments are done with high accuracy and data are interpreted with care including suitable statistics. To my point of view researchers can reproduce the given experiments based on the statements and descriptions given in the manuscript.

Answer: Thank you very much for your support to our work.

(i) According to your suggestion, we have removed the Equation 1 in the revised version.

(ii) Thank you very much for your recognition of our calculation work and the relation of the current density of the experiment and the final products, which can indeed be used for effective

scale up and commercialization; and FET devices for gaining interests in different communities, and the ribbons can be reproduce the given experiments.

② Upon reading this excellent manuscript only a few questions arose. Is the preparation mechanism of bulk black phosphorus critical or product-directing for the formation of phosphorene ribbons? There are several synthesis procedures like high pressure, complex annealing and tempering and a mineralizator based way (Sn/SnI₄ as precursor) to synthesize it. Due to the variations and the synthesis history during formation the original procedure might be of interest. I recommend to discuss the different procedures briefly in the beginning and to check the way how the black phosphorus used in this manuscript was synthesized. I believe that a different stress-strain and impurity level can directly influence the oxidation behavior in the final product. I recommend to discuss the different procedure briefly in the beginning and to check the way how the black phosphorus used in this manuscript was synthesized. I believe that a different stress-strain and impurity level can directly influence the oxidation behavior in the final product.

Answer: Many thanks for your comments.

(i) According to your suggestion, we have added a paragraph in the introduction of the preparation methods of black phosphorus, show as follows: “BP can be synthesized by different methods^{1,19,20}, from original methods involving high-pressure and high-temperature from white and red phosphorus, followed by mercury as a catalyst and the bismuth-flux methods, to the current low-pressure transport reaction routes.” It should be noted that the high quality of BP crystals of 99.998% purity are now commercially available. The bulk black phosphorus crystals we used in the experiment was bought from Nanjing XF Nano Materials TECH Co. Ltd, which was prepared via high-pressure and high-temperature method. The as-used BP in our experiment is 99.998% purity. Thus, we add a sentence in the revised version, as follows: “Commercial BP crystal prepared by a high-pressure and high-temperature method is used as the precursor of our electrochemical exfoliation method.”

(ii) We agree with that the BP crystal quality will affect the oxidation behavior in the final product, because defects are favorable sites for the oxygen adsorption. And we add a paragraph in the revised version as “Also the BP precursor crystal quality will affect the electrochemical exfoliation processes, due to defects are favorable sites for the oxygen adsorption.”

③ One major point to be readjusted is the name of the products. Due to the fact that at the edges of each ribbon P-O bonds of different kinds (P=O; P-OH) are formed there is a certain oxygen content present in each ribbon. The thinner and smaller the ribbons are the higher the ratio oxygen vs. phosphorus will be. I suggest to call the resulting species therefore phosphorene oxide nanoribbons instead of phosphorene nanoribbons in the entire manuscript.

Answer: Many thanks for your comment.

From XPS data (“The P_xO_y (including P=O, and P-O-P) is 2.25 at.% and P-POH is 7.38 at.%, respectively.” Thus, the total PO_x content is 9.9 at.%), we knew that the as-synthesized samples with slight oxygen defects, less than that of 15 at% for PO_x of phosphorene nanoribbons reported by Watts [R29](Ref 2 in our old version).

Furthermore, Favron et al. reported Ramana chemical study on exfoliated black phosphorus. This study demonstrated reported that “The statistic reveals that value of $A_g^1/A_g^2 > 0.2$ are characteristic of low oxidation levels (yellow region). The plot with open blue circles in Fig. 6e

also exhibits a trend in our samples that is probably linked to the enhanced reactivity for decreasing n. More importantly, it is shown that **pristine** samples have ratios in the range 0.4-0.6.” [R53].

Based on this study, the value of $A_g^1/A_g^2 = 0.27 \pm 0.02$ is from “phosphorene nanoribbons”, which is prepared “in an inert atmosphere”, identified to “low oxidation levels” [R29], while $A_g^1/A_g^2 = 0.37-0.42$ is from “air-stable monolayer of phosphorene”, “which is comparable to the value of the pristine monolayer phosphorene prepared in a glove box” [R54].

However, the value of A_g^1/A_g^2 from our product is ~ 0.52 (Fig. 1i), indicating it is comparable to the above value of the pristine monolayer phosphorene prepared in a glove box [R53]. Based on the above grounds, we wish to keep the product name as phosphorene nanobelts. Nevertheless, we would change the product name to phosphorene oxide nanobelts if the Reviewer considers 9.9 at.% oxidation is still quite high. Also we added a sentence in the revised version as “The value of A_g^1/A_g^2 is ~ 0.52 (Fig. 1i), indicating it is within the range of 0.4-0.6 for the pristine phosphorene prepared in a glovebox^{30,39}.” Nevertheless, we could change the product name to phosphorene oxide nanobelts if you consider 9.9 at.% is still quite high.

In sum, I recommend to publish this article in *Nature Commun.* in its present form, maybe addressing the questions I asked in the last part of my comments.

Sincerely, Prof. Dr. Tom Nilges, TU Munich

Answer: Thank you very much, for your kind recommendation!

References

- [R1] Duan, X., et al. High-performance thin-film transistors using semiconductor nanowires and nanoribbons. *Nature*, **425**, 274-278 (2003).
- [R2] Li, S., et al. Vapour-liquid-solid growth of monolayer MoS₂ nanoribbons. *Nat. Mater.* **17**, 535-542 (2018).
- [R3] Peng, H., et al. Aharonov-bohm interference in topological insulator nanoribbons. *Nat. Mater.* **9**, 225-229 (2010).
- [R4] Shin, S., Elzouka, M., Prasher, R., Chen, R. K., Far-field coherent thermal emission from polaritonic resonance in individual anisotropic nanoribbons. *Nat. Commun.*, 2019, **10**, 1-11(2019).
- [R5] Lee, S., et al. Anisotropic in-plane thermal conductivity of black phosphorus nanoribbons at temperatures higher than 100 K. *Nat. Commun.* **6**, 8573 (2015).
- [R6] Carvalho, A., et al. Phosphorene: from theory to applications. *Nat. Rev. Mater.* **1**, 1-16 (2016).
- [R7] Sun, J., et al. A phosphorene-graphene hybrid material as a high-capacity anode for sodium-ion batteries. *Nat. Nanotechnol.* **10**, 980-985 (2015).
- [R8] Y. Cai, G. Zhang, Y. Zhang, Eds., *Phosphorene: Physical Properties, Synthesis, and Fabrication*. Jenny Stanford Publishing Pte. Ltd., Singapore, 2019, **ISBN: 978-981-4774-644-2** (Hardcover), 978-0-203-71061-6 (eBook); p1.
- [R9] From the paper, Choucair, M., et al. Gram-scale production of graphene based on solvothermal synthesis and sonication. *Nature Nanotechnology* **4**, 30-33 (2009).
- [R10] Englert J. M., et al. Covalent bulk functionalization of graphene. *Nature Chemistry* **3**, 279 (2011).

- [R11] Tung, V. C., et al. High-throughput solution processing of large-scale graphene. *Nature Nanotechnology* **4**, 25 (2009).
- [R12] Zhao, T., et al. Ultrafast growth of nanocrystalline graphene films by quenching and grain-size-dependent strength and bandgap opening. *Nature Communications* **10**, 1-10 (2019).
- [R13] Seo, D. H., et al. Anti-fouling graphene-based membranes for effective water desalination. *Nature Communications* **9**, 683 (2018).
- [R14] Wu, Y., et al. The analysis of morphological distortion during AFM study of cells. *Scanning* **30**, 426-432 (2008).
- [R15] Chen, Y. et al. Research on double-probe, double- and triple-tip effects during atomic force microscope scanning. *Scanning* **26**, 155-161 (2004).
- [R16] Batmunkh, M., et al. Electrocatalytic Activity of a 2D Phosphorene-Based Heteroelectrocatalyst for Photoelectrochemical Cells. *Angew. Chem. Int. Ed.* **57**, 2644-2647 (2018).
- [R17] Bat-Erdene, M., et al. Efficiency Enhancement of Single-Walled Carbon Nanotube-Silicon Heterojunction Solar Cells Using Microwave-Exfoliated Few-Layer Black Phosphorus. *Adv. Funct. Mater.* **27**, 1704488 (2017).
- [R18] Lu, W., et al. Plasma-assisted fabrication of monolayer phosphorene and its Raman characterization. *Nano. Res.* **7**, 853–859 (2014).
- [R19] Liu, H., et al. Phosphorene: an unexplored 2D semiconductor with a high hole mobility. *ACS Nano* **8**, 4033–4041 (2014).
- [R20] Perello, D. J., Chae, S. H., Song, S., & Lee, Y. H. High-performance n-type black phosphorus transistors with type control via thickness and contact-metal engineering. *Nat. Commun.* **6**, 7809 (2015).
- [R21] Rafiee, J., et al. Wetting transparency of graphene. *Nat. Mater.* **11**, 217-222 (2012).
- [R22] Ji, S. H., et al. Atomic-scale transport in epitaxial graphene. *Nat. Mater.* **11**, 114-119 (2012).
- [R23] Tung, V., Allen, M., Yang, Y. Kaner, R. B. High-throughput solution processing of large-scale graphene. *Nat. Nanotechnol.* **4**, 25–29 (2009).
- [R24] Radisavljevic, B., Radenovic, A., Brivio, J., Giacometti, V., & Kis, A. Single-layer MoS₂ transistors. *Nat. Nanotechnol.* **6**, 147 (2011).
- [R25] Lee, Y. H., et al. Synthesis of large-area MoS₂ atomic layers with chemical vapor deposition. *Adv. Mater.* **24**, 2320-2325 (2012).
- [R26] Zhu, C., et al. Single-layer MoS₂-based nanoprobe for homogeneous detection of biomolecules. *J. Am. Chem. Soc.* **135**, 5998-6001 (2013).
- [R27] Li, L., et al. Black phosphorus field-effect transistors. *Nat. Nanotechnol.* **9**, 372–377 (2014).
- [R28] Zhang, R., Waters, J., Geim, A. K. & Grigorieva, I. V. Intercalant-independent transition temperature in superconducting black phosphorus. *Nat. Commun.* **8**, 15036 (2017).
- [R29] Watts, M. C., et al. Production of phosphorene nanoribbons. *Nature* **568**, 216-220 (2019).
- [R30] Vishnoi, P., Mazumder, M., Barua, M., Pati, S. K., & Rao, C. N. R. Phosphorene quantum dots. *Chem. Phys. Lett.* **699**, 223-228 (2018).
- [R31] Wang, X. et al. Highly anisotropic and robust excitons in monolayer black phosphorus. *Nat. Nanotechnol.* **10**, 517-521 (2015).
- [R32] Castellanos-Gomez, A., et al. Isolation and characterization of few-layer black phosphorus. *2D Mater.* **1**, 025001 (2014).
- [R33] Peng, X., & Wei, Q. Chemical scissors cut phosphorene nanostructures. *Mater. Res. Express.* **1**, 045041 (2014).
- [R34] Abate, Y. et al. Recent progress on stability and passivation of black phosphorus. *Adv. Mater.* **30**, 1704749 (2018).

- [R35] Reich, E. S., Phosphorene excites materials scientists. *Nature* **506**, 19 (2014).
- [R36] Cao, G. Z. *Nanostructures & Nanomaterials, Synthesis, Properties & Applications*, Imperial College Press, 2007.
- [R37] Rizzo, D. J., et al. Topological band engineering of graphene nanoribbons. *Nature* **560**, 204-208 (2018).
- [R38] Jacobberger, R. M., et al. Direct oriented growth of armchair graphene nanoribbons on germanium. *Nat. Commun.* **6**, 1-8 (2015).
- [R39] Li, X., Wang, X., Zhang, L., Lee, S., & Dai, H. Chemically derived, ultrasmooth graphene nanoribbon semiconductors. *Science* **319**, 1229-1232 (2008).
- [R40] Liu, Z., et al. Identification of active atomic defects in a monolayered tungsten disulphide nanoribbon. *Nat. Commun.* **2**, 1-5 (2011).
- [R41] Ziletti, A., Carvaltho, A., Campbell, D. K., Coker, D. F. & Neto, A. H. C., Oxygen defects in phosphorene. *Phys. Rev. Lett.* **114**, 046801 (2015).
- [R42] Ding, B., Chen, W., Tang, Z., & Zhang, J. Tuning phosphorene nanoribbon electronic structure through edge oxidization. *J. Phys. Chem. C.* **120**, 2149-2158 (2016).
- [R43] Hu, M., et al. Field effect transistors based on phosphorene nanoribbon with selective edge-adsorption: a first-principles study. *Physica E: Low-dimensional Systems and Nanostructures*, **98**, 60-65 (2018).
- [R44] Lei, S., Wang, H., Huang, L., Sun, Y. Y., & Zhang, S. Stacking fault enriching the electronic and transport properties of few-layer phosphorenes and black phosphorus. *Nano Lett.* **16**, 1317-1322 (2016).
- [R45] Hu, W., Lin, L., & Yang, C. Edge reconstruction in armchair phosphorene nanoribbons revealed by discontinuous Galerkin density functional theory. *Physical Chemistry Chemical Physics*, **17**, 31397-31404 (2015).
- [R46] Zhu, Z., et al. Magnetism of zigzag edge phosphorene nanoribbons. *Appl. Phys. Lett.* **105**, 113105 (2014).
- [R47] Lv, Y., et al. Highly sensitive bilayer phosphorene nanoribbon pressure sensor based on the energy gap modulation mechanism: A theoretical study. *IEEE. Eelectr. Device. L.* **38**, 1313-1316 (2017).
- [R48] Wang, L, Kutana, A, Zou, X, Yakobson, B. I., et al. Electro-mechanical anisotropy of phosphorene. *Nanoscale* **7**, 9746-9751 (2015).
- [R49] Zhou, Q., Chen, Q., Tong, Y. & Wang, J. Light-induced ambient degradation of few-layer black phosphorus: mechanism and protection. *Angew. Chem. Int. Ed.* **55**, 11437-11441 (2016).
- [R50] Liu, N., et al. Electronic and transport properties of zigzag phosphorene nanoribbons doped with ordered Si atoms. *Phys. Lett. A.* **384**, 126127 (2020).
- [R51] Lange, S, Schmidt, P, Nilges, T. Au₃SnP₇@black phosphorus: an easy access to black phosphorus. *Inorg. Chem.* **46**, 4028 (2007).]
- [R52] Köpf, M., et al. Access and in situ growth of phosphorene-precursor black phosphorus. *J. Cryst. Growth* **405**, 6-10 (2014).
- [R53] Favron, A., et al. Photooxidation and quantum confinement effects in exfoliated black phosphorus. *Nat. Mater.* **14**, 826-832 (2017).)
- [R54] Peo, J. et al. Producing air-stable monolayers of phosphorene and their defect engineering. *Nat. Commun.* **7**: 10450 (2016).

REVIEWER COMMENTS

Reviewer #1 (Remarks to the Author):

I would like to apologize to the authors for the time taken to return this review: I have had Covid19 and while now better, I was incapacitated for a number of weeks, which delayed my response and I apologise for the potential delay in publishing your work.

I appreciate your efforts in answering my queries, which for the main part you have done satisfactorily. Even though the lab closures mean that not all data can be achieved to answer my questions, given the importance of the work, I recommend that the paper published on the condition that the following two points are fully addressed:

1. The discussion is much improved, however, the following (rewritten) statement is still inaccurate: "The reported ionic scissoring method⁵ can generate high-quality, individual PNR; most (around 65%) are monolayer with POx content of 15 at.%, but the synthesized steps are time-consuming and rigorous. By contrast, our electrochemical method has simple steps; and most (around 60%) of belts are less than trilayer and lower POx content (~9.9 at.%)."

Phosphorene is well known to highly air-sensitive and the XPS taken in the Watts paper was after air-exposure for 3-5 mins see their Methods: "The POx contribution for the 'fresh' PNRs (exposed to air for 2–5 min during transfer to the spectrometer) is 15% of the total area—the same as that reported for few-layer phosphorene sheets created through liquid exfoliation methods [Nat. Commun. 6, 8563 (2015)]. Here the surface sensitivity of XPS should be highlighted, as although the PNRs that are isolated or at the surface of collections of PNR oxidize, multilayer PNRs or those buried underneath are protected—evident from the Raman response even after six days."

Thus the POx content quoted in the Watts paper is following air-exposure and not of the "as made" ribbons and in fact the data shows the ribbons are intact below the surface (despite the POx signal fully dominating after 6 days). So please modify (or remove this aspect) in the above statement in your paper to reflect this. Also you should discuss the transfer method/time of air exposure for your own XPS results – considering that your own samples may well be much purer than you previously inferred if your sample are air-exposed upon transfer.

2. I am still not convinced by the analysis of the heights. I agree that AFM of 2D materials often give an artificial height addition due to tip effects and the fact the 2D material will sit slightly proud of the surface. However, this 'extra height' will only add a single finite addition for each 2D material since subsequent layers will still have their crystallographically-defined layer spacing (unless restacked which is v unlikely for ribbons). i.e. if single layer will be 0.5 nm + X nm from measurement; 2 layer will be 1.0 nm + X nm; 3 layer will be 1.5 nm + X nm etc. Please modify your count and analysis accordingly.

Reviewer #2 (Remarks to the Author):

The authors have appropriately addressed my previous comments. I recommend publishing.

Response letter

Authors' response to all reviewers:

Below is the point-to-point response to the reviewer's comments. Authors' response are colored in light blue and modifications of the manuscript are colored in dark red.

Reviewer #1:

I would like to apologize to the authors for the time taken to return this review: I have had Covid19 and while now better, I was incapacitated for a number of weeks, which delayed my response and I apologise for the potential delay in publishing your work.

I appreciate your efforts in answering my queries, which for the main part you have done satisfactorily. Even though the lab closures mean that not all data can be achieved to answer my questions, given the importance of the work, I recommend that the paper published on the condition that the following two points are fully addressed:

Reply: Thank you very much for your comments on our work, especially after your Covid19 infection. We sincerely hope that you will recover soon! We also appreciate your kind comment "I recommend that the paper published on the condition that the following two points are fully addressed". The detailed responses to these comments are given below..

1. The discussion is much improved, however, the following (rewritten) statement is still inaccurate: "The reported ionic scissoring method⁵ can generate high-quality, individual PNR; most (around 65%) are monolayer with POx content of 15 at.%, but the synthesized steps are time-consuming and rigorous. By contrast, our electrochemical method has simple steps; and most (around 60%) of belts are less than trilayer and lower POx content (~9.9 at.%)."

Phosphorene is well known to highly air-sensitive and the XPS taken in the Watts paper was after air-exposure for 3-5 mins see their Methods: "The POx contribution for the 'fresh' PNRs (exposed to air for 2–5 min during transfer to the spectrometer) is 15% of the total area—the same as that reported for few-layer phosphorene sheets created through liquid exfoliation methods [Nat. Commun. 6, 8563 (2015)]. Here the surface sensitivity of XPS should be highlighted, as although the PNRs that are isolated or at the surface of collections of PNR oxidize, multilayer PNRs or those buried underneath are protected—evident from the Raman response even after six days."

Thus the POx content quoted in the Watts paper is following air-exposure and not of the "as made" ribbons and in fact the data shows the ribbons are intact below the surface (despite the POx signal fully dominating after 6 days). So please modify (or remove this aspect) in the above statement in your paper to reflect this. Also you should discuss the transfer method/time of air exposure for your own XPS results – considering that your own samples may well be much purer than you previously inferred if your sample are air-exposed upon transfer.

Reply 1: We really appreciate the valuable suggestions.

According to your advice, we removed part of the description of the XPS part of Watts paper in the second revised version, see page 6, lines 165-166. This sentence new reads: "The reported ionic scissoring method⁵ can generate high-quality, individual PNR; most (around 65%) are monolayer

with PO_x content of 15 at.%, but the synthesized steps are time-consuming and rigorous.”

We keep the zPNB sample protected with inert gas before the XPS test. In addition, we prepare and transfer the sample for the XPS test as quickly as possible. However, it can take about 5 minutes to transfer the sample for the XPS tests under ambient conditions, which is about the same as in Watts’ work (2–5 min), leading to additional oxidation into our XPS results. Thus, our “as-made” samples are predicted to be less oxidized than XPS samples which have been exposed to air for about 5 minutes.

According to the reviewer’s advice, the corresponding sentence in the manuscript has been revised:

① Lines 164-166 in page 6:

“By contrast, our electrochemical method has simple steps; and most (around ~~60%~~ 63%) of belts are less than ~~trilayer~~ 5 layers ~~lower~~ with a PO_x content of ~9.9 at.% (measured after exposed to air for ~5 min during transfer to the XPS spectrometer).”

② Line 383-385 in page 12:

“XPS spectra was obtained with an ESCALAB 250Xi system using Al K_α as the excitation source. During the process of transferring samples for the XPS test, the zPNBs are exposed to air for ~5 minutes, leading to some additional oxidation in the XPS results.”

2. I am still not convinced by the analysis of the heights. I agree that AFM of 2D materials often give an artificial height addition due to tip effects and the fact the 2D material will sit slightly proud of the surface. However, this ‘extra height’ will only add a single finite addition for each 2D material since subsequent layers will still have their crystallographically-defined layer spacing (unless restacked which is v unlikely for ribbons). i.e. if single layer will be 0.5 nm + X nm from measurement; 2 layer will be 1.0 nm + X nm; 3 layer will be 1.5 nm + X nm etc. Please modify your count and analysis accordingly.

Reply 2: We appreciated your comments. The thickness of monolayer phosphorene is ~0.84 nm according to previous reports [R1] and the layer spacing of phosphorene is 0.53 nm. As mentioned by the reviewer, experimental results show that the thickness of monolayer phosphorene should be (0.53 nm + x) nm. The extra height x is therefore 0.84 - 0.53 = 0.31 nm. We then used the following formula to calculate the number of layers from AFM images: Layer number = (T-0.31)/0.53, where T represents the thickness from AFM results. The updated results show that thickness of ~63% of the z-PNBs are below 5 layers, ~25% below trilayer and ~5% for monolayer.

This has been clarified in the manuscript, as follows:

① lines 116-121, in page 4:

“The statistical results indicate that the thicknesses of most of (close to ~90%) nanobelts are ~2.7 ± 1.7 nm, corresponding to ~~1-5~~ 1-8 layers (Fig. 1f and Supplementary Fig. 3), since the monolayer phosphorene defined as synthesized in the liquid phase is 0.84 nm², thus the extra height could be calculated to be 0.31 nm compared to the 0.53 nm of theoretical thickness of monolayer phosphorene. Two typical individual nanobelts with the thickness of ~2.4 nm and ~0.8 nm are shown as inset of Fig. 1f, corresponding to ~~trilayer~~ four-layer and monolayer phosphorene, respectively.”

② Lines 83-85 in page 3:

“**Fig. 1f** Thickness distribution diagram from the AFM images of 56 z-PNBs. Inset: AFM image of two typical belts onto a 300 nm SiO₂/Si substrate with the thickness of ~2.4 nm (left) and ~0.8 nm (right), corresponding to **trilayer** four-layer and monolayer phosphorene, respectively.”

Also new Figure 1f has been uploaded with changing “trilayer” to “four-layer” (inset of Figure 1f has been revised):

③ Lines 165-166 in page 6:

“By contrast, our electrochemical method has simple steps; and most (around ~~60%~~ 63%) of belts are less than **trilayer** 5 layers **lower** with a PO_x content of ~9.9 at.% (measured after exposed to air for ~5 min during transfer to the XPS spectrometer).”

Reviewer 2:

The authors have appropriately addressed my previous comments. I recommend publishing.

Reply: We are very grateful to the reviewer for supporting our work.

Reference

[R1] Sun, J., et al. A phosphorene-graphene hybrid material as a high-capacity anode for sodium-ion batteries. *Nat. Nanotechnol.* **10**, 980-985 (2015).

REVIEWERS' COMMENTS:

Reviewer #1 (Remarks to the Author):

The authors have made the necessary adjustments to the manuscript to address my comments and I now fully support publication.

Response letter

I. Response to reviewer #1:

The authors have made the necessary adjustments to the manuscript to address my comments and I now fully support publication.

Reply: Thank you very much for your comments on our work. We also appreciate your kind comment “The authors have made the necessary adjustments to the manuscript to address my comments and I now fully support publication.”.